



# TransClim (v1.0): A chemistry-climate response model for assessing the effect of mitigation strategies for road traffic on ozone

Vanessa Simone Rieger[1] and Volker Grewe[1,2]

[1]Deutsches Zentrum für Luft- und Raumfahrt, Institut für Physik der Atmosphäre, Oberpfaffenhofen, Germany
[2]also at: Delft University of Technology, Aerospace Engineering, Section Aircraft Noise and Climate Effects, Delft, Netherlands

**Correspondence:** Vanessa Simone Rieger (vanessa.rieger@dlr.de)

**Abstract.** Road traffic emits not only carbon dioxide ($CO_2$), but also other pollutants such as nitrogen oxides ($NO_x$), volatile organic compounds (VOC) and carbon monoxide (CO). These chemical species influence the atmospheric chemistry and produce ozone ($O_3$) in the troposphere. Ozone acts as a greenhouse gas and thus contributes to anthropogenic global warming. Technological trends and political decisions can help to reduce the $O_3$ effect of road traffic emissions on climate. In order

to assess the $O_3$ response of such mitigation options on climate, we developed a chemistry-climate response model called TransClim (Modelling the effect of surface *Trans*portation on *Clim*ate). It considers road traffic emissions of $NO_x$, VOC and CO and determines the $O_3$ change and its corresponding stratospheric-adjusted radiative forcing. Using a tagging method, TransClim is further able to quantify the contribution of road traffic emissions to the $O_3$ concentration. The response model bases on lookup-tables which are generated by a set of emission variation simulations performed with the global chemistry-

climate model EMAC (ECHAM5 v5.3.02, MESSy v2.53.0). Evaluating TransClim against independent EMAC simulations reveals very low deviations of all considered species (0.01 – 7 %). Hence, TransClim is able to reproduce the results of an EMAC simulation very well. Moreover, TransClim is about 6000 times faster in computing the climate effect of an emission scenario than the complex chemistry-climate model. This makes TransClim a suitable tool to efficiently assess the climate effect of a broad range of mitigation options for road traffic or to analyse uncertainty ranges by employing Monte-Carlo simulations.

## 15  1  Introduction

Mobility is getting more and more important in today's society. As residences, workplaces, schools and recreation areas are often spatially separated, there is an increasing demand on our transportation system. This leads to a steadily growing transportation volume and thus to steadily growing transportation emissions. Since 1970, emissions of greenhouse gases from transportation have more than doubled (Sims et al., 2014). In particular emissions from road traffic play a significant role. Amongst

all transportation sectors, the road traffic sector shows the largest growth rate. Emissions from this sector alone constitute more than 70 % of all greenhouse gas emissions originating from the transportation sector (Sims et al., 2014).

Road traffic emissions affect Earth's climate. Vehicles with combustion engines emit greenhouse gases such as carbon monoxide ($CO_2$) and nitrous oxide ($N_2O$). Greenhouse gases directly influence the radiation budget of the Earth and thus contribute to the anthropogenic global warming. In addition, road traffic emits also other pollutants such as nitrogen oxides





(NO$_x$), volatile organic compounds (VOC), carbon monoxide (CO), sulphur dioxide (SO$_2$) and particulate matter which also affect the atmospheric chemistry. For example, emissions of NO$_x$, VOC and CO influence the ozone (O$_3$) production and methane (CH$_4$) destruction in the troposphere. In general, road traffic emissions increase the O$_3$ concentration and reduce the atmospheric lifetime of CH$_4$ (Hoor et al., 2009). However, the process of forming and destroying O$_3$ in the troposphere is not linear. Whether O$_3$ is produced or destroyed crucially depends on the background concentrations of NO$_x$, VOC and CO. In

rural areas, additional NO$_x$ emissions usually lead to an increase of the O$_3$ concentration (so called "NO$_x$-limited" regime). But in regions with high NO$_x$ background concentrations, a further increase of NO$_x$ may even lead to a reduction of O$_3$ (so called "VOC-limited" regime, e.g. Dodge, 1977; Seinfeld and Pandis, 2006; Fowler et al., 2008).

The impact of road traffic emissions on atmospheric chemistry and on climate has already been investigated by a number of studies (e.g. Niemeier et al., 2006; Matthes et al., 2007; Fuglestvedt et al., 2008; Hoor et al., 2009; Uherek et al., 2010;

Righi et al., 2015; Mertens et al., 2018). Most studies show increasing ozone concentrations from road traffic emissions. For example at mid-latitudes, the surface concentration of O$_3$ increases by 5 – 15 % during northern hemispheric summer (Granier and Brasseur, 2003). Moreover, Reis et al. (2000) and Tagaris et al. (2015) focus on the influence of road traffic emissions on a regional scale. For example, Hendricks et al. (2018) reveal that German road traffic emissions contribute by about 0.8 % to the total anthropogenic radiative forcing. They also derive a corresponding surface temperature change of almost 5 mK (for the

year 2008).

To quantify the influence of road traffic emissions on O$_3$, most model studies apply the *perturbation method*. This method compares the results of two model simulations: one simulation with all emissions and one simulation with changed emissions. However, the perturbation method does not take non-linear relations, such as the tropospheric O$_3$ chemistry, into account (Grewe et al., 2010). Hence, it quantifies only the *impact* of road traffic emissions on O$_3$. As a variation of road traffic emissions

also affect the O$_3$ production of other emission sectors, it is important to determine the *contribution* of road traffic emissions to O$_3$ (in the following indicated with O$_3$$^{tra}$). Grewe et al. (2010) propose to apply the so-called *tagging method*. It follows the most important reaction pathways for the formation and destruction of O$_3$ and thus determines the contribution of road traffic emissions to the O$_3$ concentration. Accordingly, the perturbation method determines the *impact* and the tagging method determines the *contribution* of road traffic emissions to O$_3$. Both methods are essential to assess the total *effect* of road traffic

emissions on climate. (In the following, we use the term "effect" when referring to the impact and contribution together.) A detailed overview on the characterization and applicability of the two methods is given in table 1 of Mertens et al. (2020).

Ozone is not only harmful for the health of humans, animals and plants, it also acts as a greenhouse gas contributing to global warming. Consequently, it is crucial to reduce the effect of road traffic emissions on climate. To minimise the O$_3$ effect on climate, different mitigation options are available ranging from technical innovations to driving bans (e.g. Sims et al.,

2014). On the one hand, new technological trends such as new fuels for passenger cars, heavy goods vehicles and buses (e.g. Karavalakis et al., 2012; Suarez-Bertoa et al., 2015; Jedynska et al., 2015) change the vehicles' emissions of NO$_x$, VOC and CO and thus impact Earth's climate. On the other hand, political decisions such as financial support for electrical cars and car pooling also influence climate. Each mitigation option acts differently on O$_3$ and thus on climate. Hence, the quantification of the climate response is essential to fully assess a mitigation option.





Typically, complex chemistry-climate models are applied to assess the climate effect of traffic emissions. But these simulations are computational expensive and require a substantial amount of time. This impedes the assessment of many mitigation scenarios. Hence, we developed a new tool called TransClim (Modelling the effect of surface *Trans*portation on *Clim*ate). It is a chemistry-climate response model which efficiently determines the $O_3$ effect of a broad range of road traffic emission scenarios on climate.

Here, we present the response model TransClim and provide an assessment of the model's skills. The paper is structured as follows: In section 2, the model description of TransClim is given. The model idea, the requirements and the resulting algorithm for TransClim are described. In section 3, the calculation of the lookup-tables for TransClim is explained. The workflow of TransClim is described in section 4. Subsequently, TransClim is evaluated against simulations with the global chemistry-climate model EMAC in section 5. Section 6 gives an overall assessment of the response model.

The work presented in this paper bases on the PhD thesis by V. S. Rieger. Hence, significant parts of the text already appeared in Rieger (2018).

## 2   Model description of TransClim

### 2.1   Model idea

## Model concept of TransClim

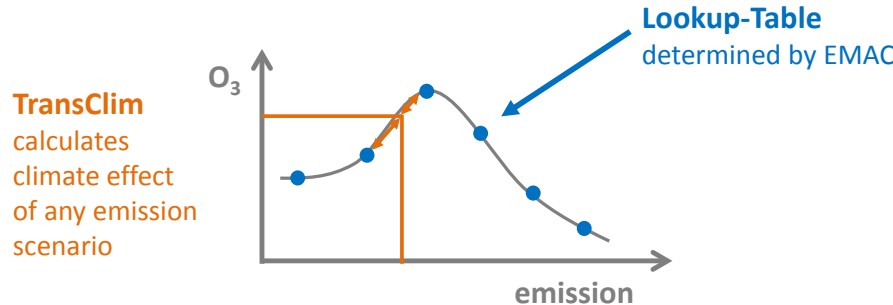

**Figure 1.** TransClim is based on lookup-tables (LUT) determined from simulations with the global chemistry-climate model EMAC. By interpolating within the LUTs, TransClim determines the $O_3$ concentration change and the corresponding climate effect of any road traffic emission scenario covered by the LUTs.

To reduce the $O_3$ effect of road traffic emissions on climate, mitigation strategies need to be developed and evaluated by
assessing their expected climate effect. The new tool TransClim is a chemistry-climate response model which efficiently assess the $O_3$ effect of a change in road traffic emissions on climate.





To quickly determine the climate effect of a given emission scenario, TransClim does not explicitly calculate the chemical and physical processes. Instead, it uses lookup-tables (LUT) which contain pre-calculated relations between emissions and their climate effects. Figure 1 shows the basic principle of TransClim for the example of tropospheric $O_3$. Road traffic emissions of $NO_x$, VOC and CO are varied and the corresponding tropospheric $O_3$ change is simulated with the global chemistry-climate model EMAC (see details in sect. 3.2.1). Note that the relation between the emission variation and the $O_3$ change is non-linear as the $O_3$ chemistry in the troposphere is non-linear. The ratio of VOC/$NO_x$ and CO/$NO_x$ concentration in the atmosphere crucially determines whether $O_3$ is formed or destroyed (e.g. Seinfeld and Pandis (2006)). These relations between emission variation and $O_3$ change are used to create lookup-tables (LUT) for TransClim. TransClim interpolates within these LUTs and determines the climate effect (e.g. $O_3$ change) of a specific road traffic emission scenario.

In this manner, TransClim can not only determine the climate effect of $O_3$ but also for other variables such as the hydroxyl radical (OH) or the stratospheric-adjusted radiative flux change at top of the atmosphere. Hence, it enables to quickly assess the climate effect of a broad range of emission scenarios.

## 2.2 Requirements

The aim of TransClim is to assess the effect of road traffic emissions of $NO_x$, VOC and CO on tropospheric $O_3$ and its respective effect on climate (such as radiative forcing). Thus, the algorithm of TransClim, which combines pre-calculated relations between emissions and climate effect, needs to meet the following requirements:

1. Road traffic emissions of $NO_x$, VOC and CO are considered by the algorithm. These key species are involved in the formation of $O_3$ in the troposphere. Thus, the algorithm is able to quantify the resulting total change in $O_3$ concentration as well as the contribution of road traffic emissions to the $O_3$ concentration ($O_3^{tra}$, derived by the tagging method, see appendix A).

2. The non-linearity of the tropospheric $O_3$ chemistry is considered.

3. Road traffic emissions from different emission regions (e.g. Europe, Germany, North America, ...) are regarded. Within each of these emission regions, the road traffic emissions are varied.

4. The algorithm determines the geographical pattern of the $O_3$ and $O_3^{tra}$ change resulting from a given road traffic emission scenario. This allows for assessing not only the global but also the regional effects as the downwind effect can differ from the source region effect.

5. The radiative forcing of $O_3$ and $O_3^{tra}$ are calculated.

6. The background concentration of $O_3$ is taken into account. This allows for considering different future emission scenarios such as the Representative Concentration Pathways RCP (van Vuuren et al., 2011). The $O_3$ background concentration can vary in a future emission scenario and thus also the climate effect of the road traffic emissions changes.





7. The algorithm is computational very efficient. This means that the climate effect of a given emission scenario is calculated within minutes or hours. Differences in the results compared with complex chemistry-climate model simulations generally remain below 10 %.

## 2.3 Algorithm

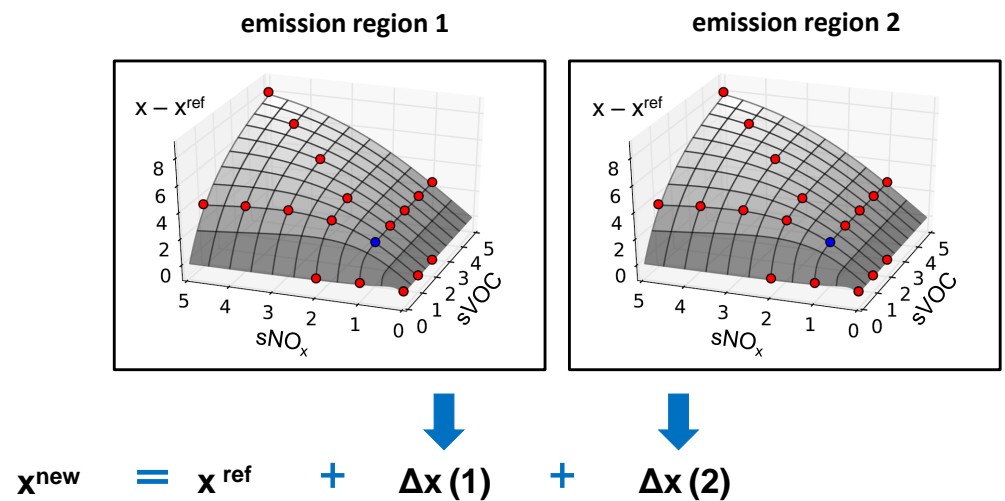

**Figure 2.** Sketch of interpolation algorithm used by TransClim. For each emission region, a LUT contains the change in variable $x$ ($x - x^{\text{ref}}$) and the emission scaling factors for $NO_x$, VOC and CO emissions ($sNOx$, $sVOC$, $sCO$). In the figure, only $sNOx$ and $sVOC$ are displayed. The blue dot indicates the reference simulation ($sNOx = 1, sVOC = 1$). The red dots indicate the emission variation simulations performed with EMAC. After linearly interpolating within the LUT for each emission region $i$, the resulting changes $\Delta x(i)$ are added to reference $x^{ref}$. This procedure is performed for every grid box or for tropospheric or global means.

After testing several algorithms, the following algorithm was identified to produce very good results (Rieger, 2018). For the sake of clarity, fig. 2 shows a sketch of the algorithm for only two emission regions (e.g. Western and Eastern Europe) and for only two road traffic emission species $NO_x$ and VOC. For each emission region, a separate LUT is created: *emission scaling*

*factors* for $NO_x$, VOC and CO road traffic emissions ($sNOx$, $sVOC$, $sCO$), which describe the factors by which the reference emissions are scaled, are used as input variables. Thus, each LUT has three dimensions (in fig. 2, two dimensions). The LUT then provides the change ($\Delta x$) of variable $x$ with respect to the reference simulation ($x^{ref}$):

$$\Delta x(i) = x(i) - x^{\text{ref}}(i) = \text{LUT}(\ sNOx(i),\ sVOC(i),\ sCO(i)\ )$$

Consequently, each output variable has its own LUT.

To obtain the desired variable $x^{new}$ for a given road traffic emission scenario, the corresponding emission scaling factors ($sNOx$, $sVOC$, $sCO$) for each emission region $i$ are used as input and the change $\Delta x(i)$ for each emission region is calculated





by linearly interpolating within the respective LUT. Since for example an emission change of $NO_x$ in one emission region affects also the $O_3$ concentration in an emission region which is far away from the source emission region, it is important to consider the effect of all emission regions together. Thus, the computed $\Delta x(i)$ of each emission region $i$ is added to the reference $x^{ref}$ (see fig. 2):

$$x^{new} = x^{ref} + \sum_i \Delta x(i) \tag{1}$$

This method can be performed either for the tropospheric or global mean of a variable $x$ or for all grid boxes of a global climate simulation. Thus, it applies for 1-dimensional variables, such as global radiative forcing, as well as for multi-dimensional variables, such as the $O_3$ concentration.

This approach offers a fast method to estimate the effect of road traffic emissions on e.g. tropospheric $O_3$. Using a standard computer, it takes 0.2 s to compute the global mean climate effect of an emission scenario in one emission region. To calculate a three-dimensional variable, e.g. the new $O_3$ concentration in the whole atmosphere, for an emission scenario, it takes about 15 min. In this case, the algorithm is applied to each grid box of a global climate simulation: to 64 latitudes, 128 longitudes and 90 vertical pressure levels (this is the resolution of the global climate-chemistry model EMAC used to generate the LUTs, see sect. 3).

## 3 Calculation of lookup-tables

### 3.1 Emission regions

To determine the effect of road traffic emissions from different emission regions, eleven emission regions are defined (fig. 3): Germany, Western Europe, Northern Europe, Eastern Europe, Southern Europe, North America, South America, China, India, Southeast Asia and Japan/South Korea. The emission region Western Europe contains most of France, Great Britain and Ireland. Scandinavia is named Northern Europe. Eastern Europe consists of not only the Eastern European countries but also some parts of the Balkan countries: Slovenia, Croatia, Romania and the northern part of Bosnia and Herzegovina and Serbia. The emission region Southern Europe contains the whole European Mediterranean such as Iberian Peninsula, Italy, the Southern Baltic countries, Greece, Cyprus and the Western Turkey. The region North America merges USA, Canada, Northern Mexico and Cuba. The emission region Asia is divided into China, India, Southeast Asia and one emission region containing Japan and South Korea.

Table 1 gives the total amounts of road traffic emissions for $NO_x$, CO and VOC in the eleven emission regions, the remaining part of the world and the global values as derived from the emission inventory MACCity (Granier et al., 2011). The emission region Germany has low VOC road traffic emissions of only 0.09 Tg(C) yr$^{-1}$ compared to the other European emission regions. Eastern and Southern Europe show high CO road traffic emissions of about 4 Tg(CO) yr$^{-1}$. In general, the emission regions China, India, Southeast Asia as well as North and South America show high road traffic emissions. The global road traffic emissions for $NO_x$ are 20.31 Tg(NO) yr$^{-1}$, for CO 145.80 Tg(CO) yr$^{-1}$ and for VOC 17.22 Tg(C) yr$^{-1}$.



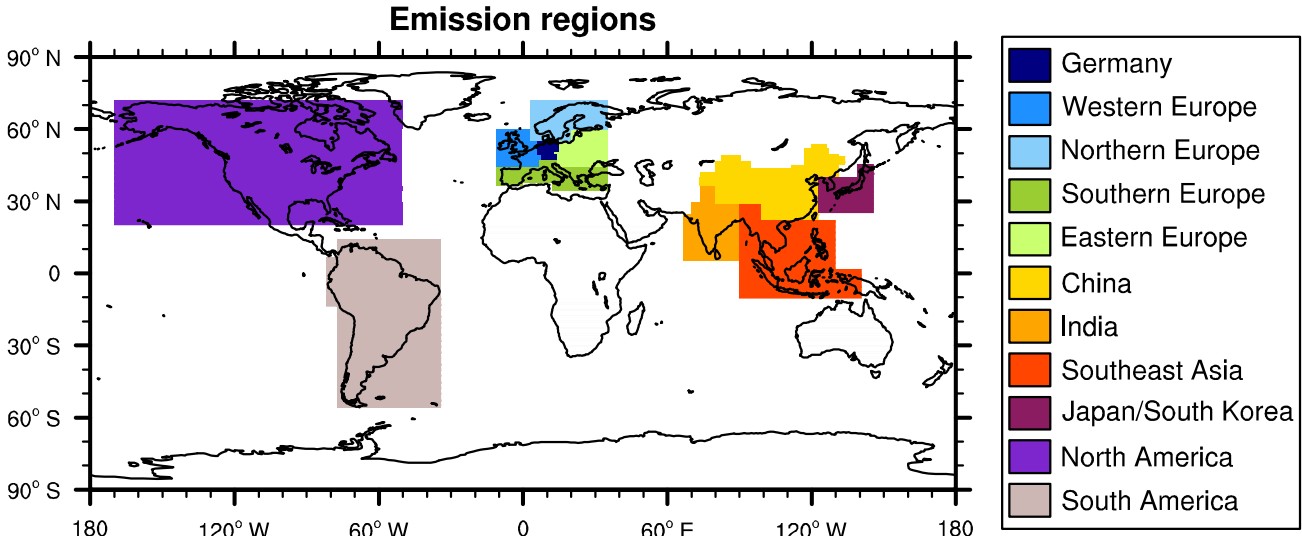

**Figure 3.** Eleven emission regions which are defined for the LUTs of TransClim.

Russia, Africa, Arabian Peninsula and Australia are not regarded as a separate emission region yet. However, further regions can be easily considered by expanding the LUTs.

## 3.2 Emission variation simulations with the global chemistry-climate model EMAC

### 3.2.1 Model description of EMAC

Here, we use the global chemistry-climate model ECHAM/MESSy Atmospheric Chemistry (EMAC) to generate the LUTs for TransClim. EMAC is a numerical chemistry and climate simulation system that includes sub-models describing tropospheric and middle atmosphere processes and their interaction with oceans, land and human influences (Jöckel et al., 2010). It uses the second version of the Modular Earth Submodel System (MESSy2) to link multi-institutional computer codes. The core atmospheric model is the 5th generation European Centre Hamburg general circulation model (ECHAM5, Roeckner et al., 2006). For the present study, we applied EMAC (ECHAM5 version 5.3.02, MESSy version 2.53.0) in the T42L90MA-resolution, i.e. with a spherical truncation of T42 (corresponding to a quadratic Gaussian grid of approx. 2.8 by 2.8 degrees in latitude and longitude) with 90 vertical hybrid pressure levels up to 0.01 hPa. The applied model setup is similar to the model setup of the EMAC simulation *RC1SD-base-10a* described in detail in Jöckel et al. (2016). In the following, the most important configuration features of the simulation are summarized. The simulation is free running and has a time step length of 12 minutes.

The chemical mechanism is solved by the submodel MECCA (Module Efficiently Calculating the Chemistry of the Atmosphere, Jöckel et al. (2010); Sander et al. (2011)) which regards the basic chemistry of the tropo- and stratosphere. It considers 188 chemical species interacting in 218 gas phase, 12 heterogeneous and 68 photolysis reactions.




| | NO$_x$ | CO | VOC | VOC |
|---|---|---|---|---|
| | Tg(NO) yr$^{-1}$ | Tg(CO) yr$^{-1}$ | Tg(C) yr$^{-1}$ | Tg(VOC) yr$^{-1}$ |
| Germany | 0.486 | 1.148 | 0.090 | 0.117 |
| Western Europe | 0.730 | 2.331 | 0.205 | 0.267 |
| Northern Europe | 0.342 | 0.831 | 0.167 | 0.218 |
| Eastern Europe | 0.561 | 4.246 | 0.408 | 0.532 |
| Southern Europe | 0.840 | 4.050 | 0.430 | 0.561 |
| China | 2.258 | 16.854 | 3.649 | 4.760 |
| India | 1.562 | 9.050 | 0.840 | 1.096 |
| Southeast Asia | 1.094 | 8.102 | 2.919 | 3.807 |
| Japan / South Korea | 0.728 | 2.910 | 0.903 | 1.178 |
| North America | 4.473 | 35.829 | 1.276 | 1.664 |
| South America | 1.946 | 13.825 | 1.877 | 2.448 |
| Rest of the world | 5.291 | 46.622 | 4.459 | 5.816 |
| GLOBAL | 20.311 | 145.798 | 17.223 | 22.463 |

**Table 1.** Road traffic emissions per emission region for the year 2010 derived from the emission inventory MACCity (Granier et al., 2011). Global emissions are given in the last row.

To detect small perturbations (such as variations in emissions of road traffic), we apply the Quasi Chemistry Transport Model (QCTM) mode for EMAC (Deckert et al., 2011). It decouples the chemistry from the dynamics by prescribing climatologies for the radiation calculation and the hydrological cycle.

The radiative fluxes are computed by the submodel RAD (Dietmüller et al., 2016). The longwave radiative spectrum is divided into 16 spectral bands (Mlawer et al., 1997). The shortwave radiative spectrum consists of 4 spectral bands in the
175 troposphere and up to 55 bands in the stratosphere and mesosphere (Fouquart and Bonnel, 1980; Nissen et al., 2007).

As EMAC is run in the QCTM mode, the calculation of the radiative fluxes, which feed back to the model simulation, is based on climatologies of $CO_2$, $CH_4$, $O_3$, $N_2O$, $CF_2Cl_2$ and $CFCl_3$ (first call of the radiation module, rad01). To further determine the radiative forcing of $O_3^{tra}$, additional radiative fluxes have to be calculated. Firstly, the radiative fluxes of the $O_3$ field which is modified by the model chemistry (provided by the submodel MECCA) are calculated (second call of the radiation
module, rad02). Secondly, the radiative fluxes of $(O_3 - O_3^{tra})$ are computed (third call of the radiation module, rad03). In a post processing step, the radiation fluxes calculated by the second and third call of the radiation module are subtracted from each other (rad02 − rad03) to obtain the radiative fluxes caused by $O_3^{tra}$ (Mertens et al., 2018). For both additional calls of the radiation module (rad02, rad03), the stratospheric-adjusted radiative fluxes are computed.

Anthropogenic emissions such as emissions from road traffic, ships, aviation, industry, agricultural waste burning and
185 biomass burning are provided by the MACCity emission inventory (Granier et al., 2011). The submodel ONEMIS (Kerkweg et al., 2006) computes emissions during the simulation (i.e. online) such as emissions of soil NO$_x$ (following Yienger





and Levy, 1995) and biogenic isoprene ($C_5H_8$) emissions (following Guenther et al., 1995). For $NO_x$ from lightning, the parameterization of Grewe et al. (2001) is applied with lightning $NO_x$ emissions scaled to approx. 5 Tg(N) per year.

To specify the contribution of road traffic emissions to the $O_3$ concentration, the submodel TAGGING is used. It applies the tagging method briefly described in appendix A. A detailed description can be found in Grewe et al. (2010), Grewe et al. (2017) and Rieger et al. (2018).

The time period of July 2009 to December 2010 is simulated. The first half year is taken as spin-up period, the remaining year is used for the analysis. Due to limited computational resources, it is only possible to use one year for the analysis. An EMAC simulation performed for a time period of three years shows that the year-to-year variability of tropospheric $O_3$ and $O_3^{tra}$ is quite low which allows for using only one year for the analysis (see also Hoor et al. (2009)).

### 3.2.2 Setup of EMAC emission variation simulations

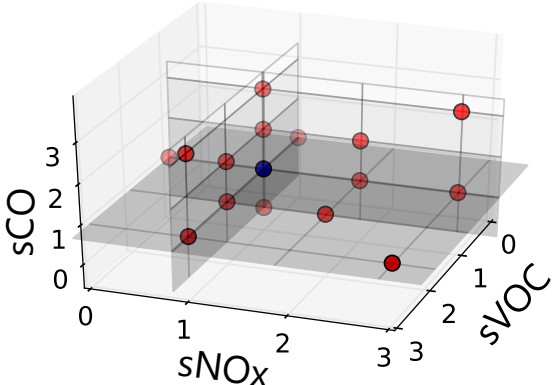

**Figure 4.** Sketch of emission variation of $NO_x$, VOC and CO road traffic emissions for each emission region. The emission scaling factors sNOx, sVOC and sCO are shown. Each dot presents a simulation with EMAC. The reference simulation is displayed by a blue dot and has emission scaling factors for $NO_x$, VOC and CO of 1.

To generate the LUTs, emission variation simulations are performed with EMAC. In each of the eleven emission regions (see sect. 3.1), the emission scaling factors sNOx, sVOC and sCO are varied separately. The emission scaling factors are defined in the MESSy run script for each emission region. The respective EMAC output for the year 2010 is used as input for the LUTs.

First of all, a reference simulation is performed with all emission scaling factors $(sNOx, sVOC, sCO)$ in all emission regions set to 1. Then, $sNOx, sVOC$ and $sCO$ are changed in one of the eleven emission regions while the factors of the remaining emission regions are kept constant at 1. For one emission region, the sketch in fig. 4 presents the principle of emission variation simulations. As it is computationally too expensive to cover the whole domain of possible emission variation of $NO_x$, VOC and CO, only two emission scaling factors are varied at the same time. The third factor is left at 1. For the LUTs, emission variation simulations with EMAC are performed using emission scaling factors varied between 0 (corresponds to no emissions) and 2 (corresponding to a duplication of emissions) in each emission region. Additionally, two emission variation simulations





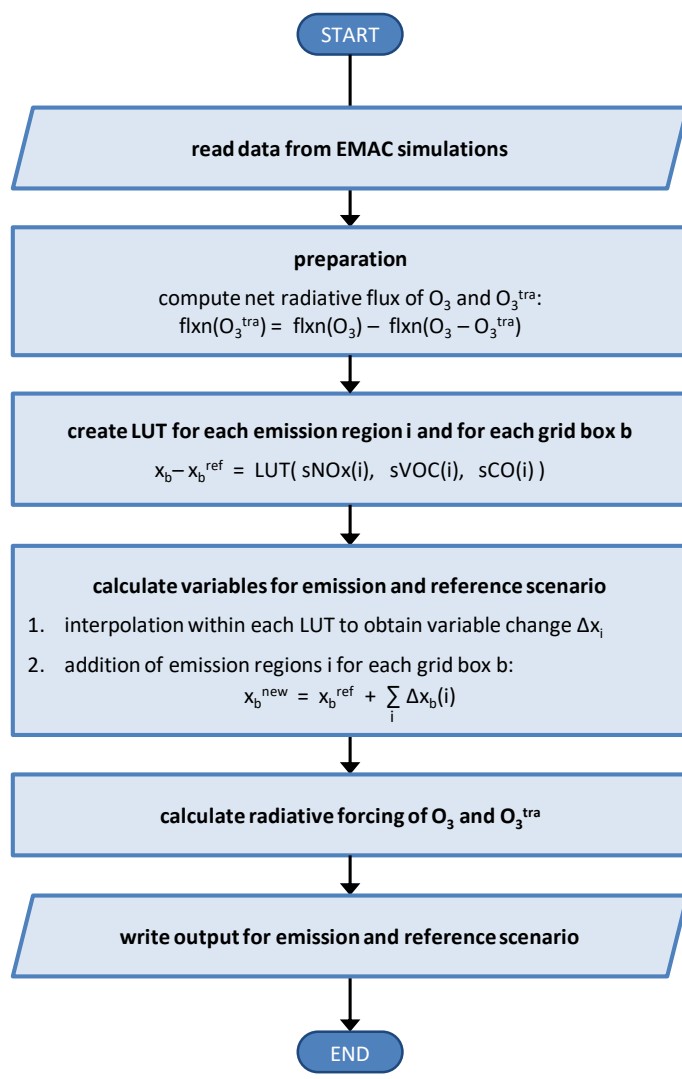

**Figure 5.** Workflow of TransClim showing the main calculation steps. For a given emission and reference scenario, TransClim computes the resulting climate effect such as the stratospheric-adjusted radiative forcing at top of the atmosphere.

with $sNOx, sVOC$ and $sCO$ set all to 0 and 0.5 in each emission region are conducted. In the supplement, table S3 shows a list of all emission variation simulations performed with EMAC which are currently available.





## 4  Workflow of TransClim

To perform a simulation with TransClim, it is necessary to define an emission scenario as well as a reference scenario to which the emission scenario is compared to. This is important when determining the climate effect such as the radiative forcing of the emission scenario.

Fig. 5 shows the workflow of the main calculation steps. First of all, all required input data from the emission variation simulations performed with EMAC (see sect. 3.2) are read (see table S5 in the supplement). To prepare a simulation, TransClim

computes the net radiative fluxes of $O_3$ and $O_3^{tra}$ from the second and third call of the radiation module (see sect. 3.2.1) by summing up the shortwave and longwave fluxes. These radiative fluxes correspond to the ozone concentration (flxn($O_3$)) and to the difference of the ozone concentration and the ozone contribution (flxn($O_3 - O_3^{tra}$)) which are modified by the emission variation. Subsequently, the net radiative fluxes due to $O_3^{tra}$ is determined by subtracting flxn($O_3 - O_3^{tra}$) from flxn($O_3$).

In the following step, TransClim creates LUTs with the dimensions $sNOx$, $sVOC$ and $sCO$ representing the variable

change $x - x^{ref}$ for each emission region and each grid box. For example, for the variable $O_3$, TransClim generates 8.110.080 LUTs for 11 emission regions, for 90 levels, for 64 latitudes and for 128 longitudes. For the tropospheric mean of $CH_4$ lifetime, only 11 LUTs for the 11 emission regions are produced.

Afterwards, TransClim computes the variables for a given emission and reference scenario applying the algorithm described in sect. 2.3. As a first step, the algorithm linearly interpolates in the corresponding LUT for each variable $x$ of each emission

region $i$ and grid box $b$ to obtain the difference: $\Delta x_b(i) = x(i) - x^{ref}(i)$. In a second step, the results of each emission region are added to the value of the reference EMAC simulation: $x_b^{new} = x_b^{ref} + \sum_{i=1}^{n} \Delta x_b(i)$, with $n$ being the number of emission regions (here $n = 11$). For multi-dimensional variables, this procedure is repeated for all grid boxes $b$ (levels, latitudes and longitudes).

Subsequently, the radiative forcings for $O_3$ and $O_3^{tra}$ are computed by subtracting the radiative fluxes of the reference

EMAC simulation from the radiative fluxes of the emission and reference scenario calculated by TransClim. Additionally, the net stratospheric-adjusted radiative forcing at top of the atmosphere of the emission scenario towards the reference scenario is determined. In a final step, the interpolated values for the emission and reference scenario are written to netCDF files.

## 5  Model evaluation

In the following section, the model TransClim is evaluated for two cases: Firstly, TransClim is compared with an equivalent

EMAC simulation. Secondly, TransClim is evaluated against other EMAC simulations performed within the DLR project VEU1 (Verkehrsentwicklung und Umwelt 1, i.e. Transport and the Environment 1, www.dlr.de/VEU; Hendricks et al., 2018).

### 5.1  Comparison with equivalent EMAC simulation

In this section, a TransClim simulation is compared with an equivalent EMAC simulation. Based on the set of emission variation simulations which are currently available for the LUTs of TransClim (see table S3 in supplement), a set of emission scaling





factors for each emission region is chosen in such a way that a broad range of emission variation is given. For this evaluation simulation, the following emission scaling factors are applied: in Germany, only $NO_x$ emissions are reduced. In Western and Northern Europe, two of the emission types are varied simultaneously. In contrast, all emissions are enhanced or lowered by the same factor in the regions Eastern and Southern Europe. The emissions of the remaining regions are not changed. The values for the emission scaling factors are summarized in table 2. The road traffic emissions of $NO_x$, VOC and CO are only changed in Europe to test if the algorithm of TransClim also works on a regional scale. Here, the non-linearities of the $O_3$ chemistry are expected to be larger than on global scale. Hence, this scenario with a large variation of emissions in Europe is expected to be a difficult test case for TransClim.

| Emission region | Emission scaling | | |
| --- | --- | --- | --- |
| | sNOx | sVOC | sCO |
| Germany | 0.3 | 1.0 | 1.0 |
| Western Europe | 0.1 | 1.0 | 0.9 |
| Northern Europe | 1.6 | 0.7 | 1.0 |
| Eastern Europe | 1.3 | 1.3 | 1.3 |
| Southern Europe | 0.5 | 0.5 | 0.5 |

**Table 2.** Emission scaling factors for the comparison of a TransClim simulation with an equivalent EMAC simulation. The remaining emission regions that are not listed in this table are kept constant at 1.

For the comparison, the emission scaling factors listed in table 2 are used for a simulation with EMAC (see sect. 3.2.1) and for a simulation with TransClim (based on the LUTs as described in sect. 3.2.2). Fig. 6 shows the results for ozone ($O_3$) and the contribution of road traffic emissions to ozone ($O_3^{tra}$) over Europe from the TransClim simulation and the relative difference to the equivalent EMAC simulation. The tropospheric $O_3$ and $O_3^{tra}$ columns (in Dobson units) are shown. At lower latitudes, photolysis rates are generally larger producing more $O_3$. The relative differences between the TransClim and EMAC simulation are very low, i.e. for $O_3$ the maximum deviations are below 0.01 % and for $O_3^{tra}$ below 0.3 %. Throughout most of the domain, TransClim underestimates $O_3$ and $O_3^{tra}$ compared to EMAC (the reason for this is explained below). Only over the Mediterranean countries, TransClim computes slightly larger values than EMAC.

The relative differences in ozone ($O_3$), hydroxyl radical (OH) and net flux at top of the atmosphere ($flxn(O_3)$) as well as the corresponding contributions of road traffic emissions ($O_3^{tra}$, $OH^{tra}$, $flxn(O_3^{tra})$) obtained by TransClim in comparison to EMAC are shown in fig. 7. For the tropospheric $O_3$ column, the largest deviations of -0.009 % are found in Northern Europe and span over the Northern Hemisphere. Deviations of up to 0.1 % in tropospheric mean of OH are only found over Europe. For $flxn(O_3)$, the relative differences between EMAC and TransClim are very small (in average < 0.001%). The contributions of road traffic emissions ($O_3^{tra}$, $OH^{tra}$ and $flxn(O_3^{tra})$) show larger differences of up to -7 % in the Southern Hemisphere. However, the contributions of road traffic emissions in the Southern Hemisphere are generally very small. To compute the

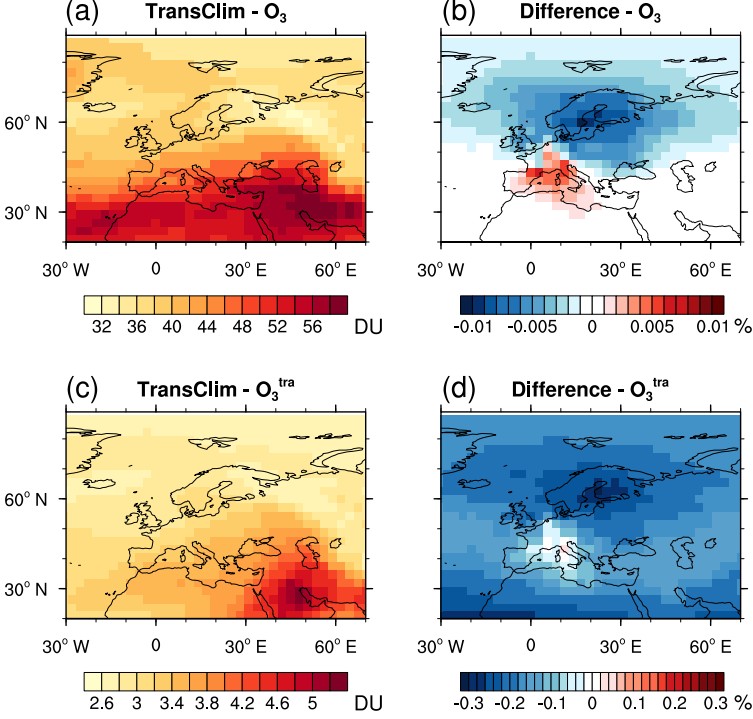

**Figure 6.** Comparison between TransClim and an equivalent EMAC simulation. Tropospheric $O_3$ (a) and $O_3^{tra}$ (c) columns are given in Dobson units (DU). The relative differences of the TransClim results with respect to the EMAC simulation are shown for the tropospheric columns ((b) and (d)).

relative differences, the absolute differences are divided by these small values in the Southern Hemisphere. The noise generate by this calculation is responsible for the relatively large differences in this region.

Additionally, the frequency distributions of the relative differences are displayed in appendix B. Although a few grid boxes show large deviations, the deviations typically found in the troposphere are rather small (below 7 %). These comparisons show that the deviations between the EMAC and TransClim simulation are generally very small.

Fig. 8 shows a sketch of the interpolation error in the results calculated by TransClim. Blue dots indicate the LUT values for $O_3^{tra}$ depending on the $NO_x$ emission scaling factors in Germany. The blue line presents the non-linear relationship between

the $NO_x$ emissions and $O_3^{tra}$. The interpolation algorithm of TransClim is implemented in Python. The LUTs of TransClim are 3-dimensional and the data is arranged on an irregular grid. For an interpolation in a multi-dimensional irregular data structure, the library SciPy in Python offers only the option to interpolate linearly within this grid. The curvature of the non-linear relationship between $NO_x$ emissions and $O_3^{tra}$ is negative. Thus, a linear interpolation within the LUT (indicated by the black line) causes an underestimation of the interpolated value. The error which is caused by the linear interpolation is indicated with

the red line. However, the resulting errors are so small (see fig. 6 and fig. 7) that the application of a linear interpolation is justified.





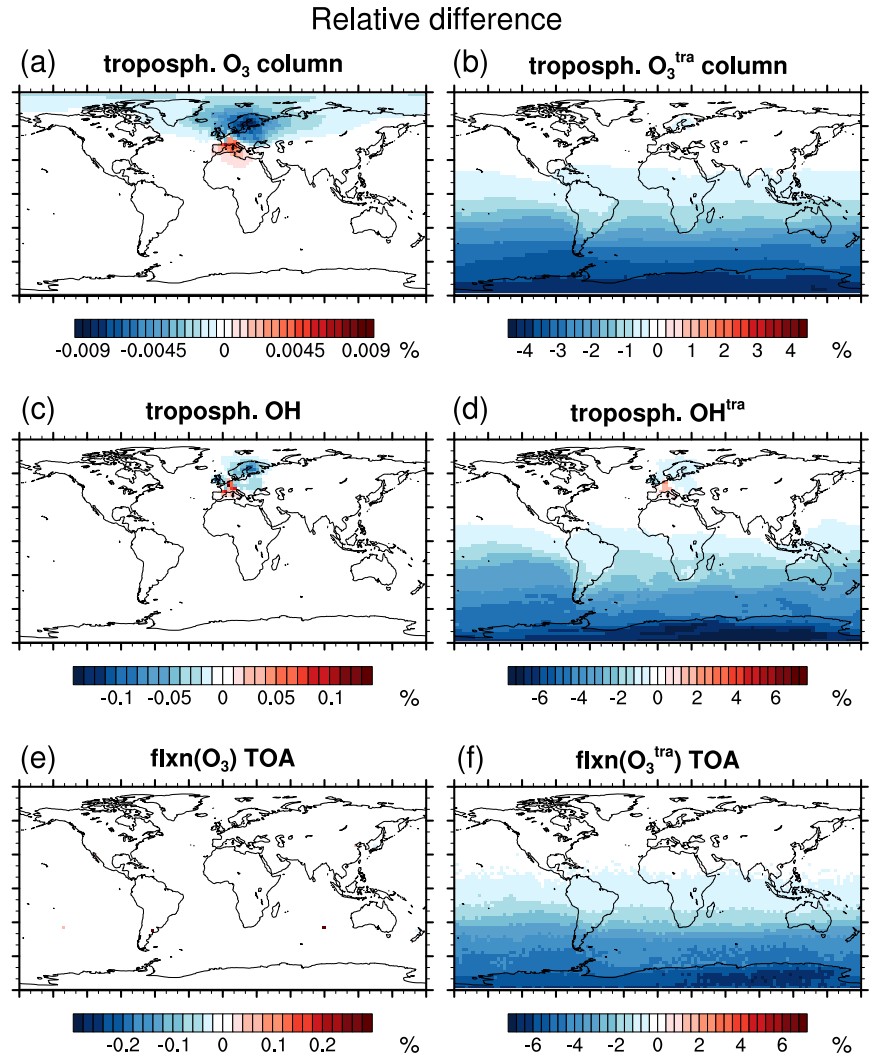

**Figure 7.** Relative difference between TransClim and EMAC simulation. Ozone ($O_3$), hydroxyl radical (OH) and ozone net radiative fluxes (flxn($O_3$)) as well as the contribution to ozone ($O_3^{tra}$), to hydroxyl radical ($OH^{tra}$) and to ozone net radiative fluxes (flxn($O_3^{tra}$)) are shown. For $O_3$ and $O_3^{tra}$, the relative difference of the tropospheric columns are shown ((a) and (b)). For OH and $OH^{tra}$, the deviations of the tropospheric means are displayed ((c) and (d)). The values at top of the atmosphere (TOA) are shown for flxn($O_3$) and flxn($O_3^{tra}$) ((e) and (f)).

Summing up, despite the general slight underestimation of TransClim, the deviations between the results obtained by TransClim and EMAC are very low (below 7 %). This shows that TransClim reproduces this EMAC simulation very well.

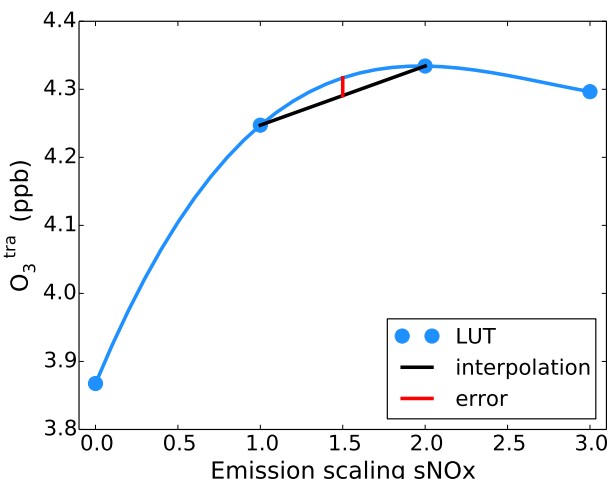

**Figure 8.** Sketch of the interpolation error caused by the linear interpolation in the LUT

## 5.2 Comparison with VEU1 simulations

In this section, EMAC simulations performed within the project VEU1 are reproduced with TransClim to assess the performance of TransClim. The DLR project VEU1 (Verkehrsentwicklung und Umwelt 1, i.e. Transport and the Environment 1, Henning et al. (2015), www.dlr.de/VEU) examined the German transport and its effect on the environment (Hendricks et al., 2018). In VEU1, EMAC simulations were performed to quantify the climate impact of future road traffic emission scenarios. Road traffic emissions for the year 2030 were determined and their impact $NO_x$, $O_3$ and OH was computed with EMAC. This

offers a good opportunity to test the performance of TransClim.

Within the scope of the project VEU1, German road traffic emissions were derived for present day conditions as well as for possible future scenarios. Based on socio-economic data such as population, households, income levels, economic development and demographic trends, the transport demand was determined. To compute the emissions from road traffic, railways and inland shipping as well as passenger and freight transport were regarded. For the passenger transport, different transport modes such as

motorised private transport, public transport, bicycles and pedestrians were taken into account. Additionally, different vehicle and fuel types as well as the emission classes were considered. The development of new technologies in the transport sectors were modelled as well. Considering all these different factors, a *baseline emission scenario* for German road traffic emissions for the years 2008, 2020 and 2030 was created.

In VEU1, the climate impact of this baseline emission scenario was simulated with EMAC only for the year 2030 by using

the perturbation method. This method compares two EMAC simulations: one simulation contains all emissions and another simulation neglects the road traffic emissions. In order to obtain a robust signal of the German road traffic emissions, the perturbation signal was enhanced. Thus, not only the road traffic emissions in Germany but the road traffic emissions in all European countries were set to zero. This method determines the climate impact of the European road traffic emissions.





Subsequently, the resulting European radiative forcing from the change in $O_3$ was in turn downscaled to estimate the $O_3$
radiative forcing of the German road traffic emissions. Additionally, the $CH_4$ lifetime change caused by German road traffic
emissions was deduced from the EMAC simulation. More details on the specific model setup of the EMAC simulations are
found in Gottschaldt et al. (2013) and Hendricks et al. (2018).

The results obtained by the project VEU1 offer the opportunity to evaluate TransClim with respect to the climate impact of
$O_3$ and $CH_4$ lifetime change caused by regional transport emissions. TransClim uses the road traffic emissions of VEU1 and is
then used to reproduce the results from the EMAC simulations performed in VEU1. The emission scaling factors (factors by
which the reference emissions are scaled) for TransClim are presented in table 3. For this simulation, the resulting $NO_x$ and
OH mixing ratios are also computed by TransClim.

| Emission region | Emission scaling | | | year |
|---|---|---|---|---|
| | sNOx | sVOC | sCO | |
| Germany | 1.136 | 1.509 | 1.032 | 2008 |
| Germany | 0.514 | 0.802 | 0.422 | 2020 |
| Germany | 0.298 | 0.724 | 0.382 | 2030 |
| Western Europe | 0.729 | 0.462 | 0.490 | |
| Northern Europe | 0.379 | 0.305 | 0.723 | |
| Eastern Europe | 0.677 | 0.415 | 0.366 | |
| Southern Europe | 0.725 | 1.388 | 0.521 | |

**Table 3.** Emission scaling factors for the TransClim simulation to reproduce the VEU1 simulations with EMAC. The emission scaling factors
in Germany for the years 2008, 2020 and 2030 are also indicated. For the remaining European regions, the emission scaling factors are set
constant for the years 2008, 2020 and 2030. The remaining emission regions are not listed in the table as they are kept at 1.

The change in the zonal means of $NO_x$, $O_3$ and OH caused by the European road traffic emissions (i.e. difference between
the "reference simulation" and "no European road traffic simulation") for the year 2030 are shown in fig. 9. The first and
second column show the relative and absolute change derived from TransClim. The third column presents the absolute changes
obtained with EMAC in VEU1 (Hendricks et al., 2018). European road traffic emissions increase $NO_x$ over the Northern
Hemisphere. The increase (up to 4 %) is very confined to the latitudes where the European road traffic emissions occur.
Furthermore, European road traffic emissions increase $O_3$ in the Northern Hemisphere. The $O_3$ rise is not only bound to the
lower troposphere but reaches high up to the tropopause region. It even stretches into the lower stratosphere where $O_3$ from
European road traffic emissions is found over the tropics. The zonal mean is increased by up to 0.5 % in the Northern lower
troposphere. Moreover, European road traffic emissions cause an OH increase in the lower troposphere which is rather confined
to the emission region. It further decreases OH in the upper troposphere. TransClim reproduces the patterns of $NO_x$ and $O_3$
increases very well. However, TransClim underestimates the OH increase caused by European road traffic emissions. In VEU1,
the OH increase reaches the tropopause region in the Northern Hemisphere. In contrast, TransClim confines the OH increase



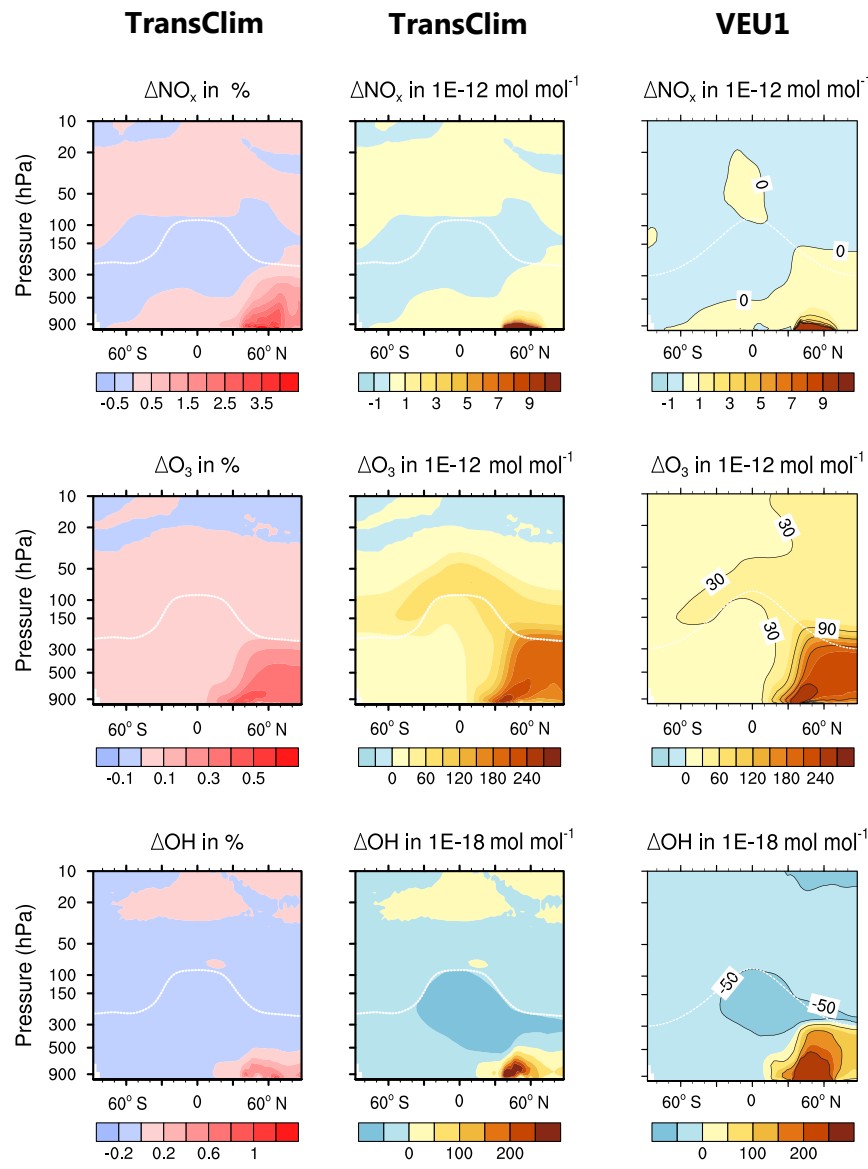

**Figure 9.** Zonal mean of relative and absolute $NO_x$, $O_3$ and OH change caused by European road traffic emissions for the year 2030. Simulations performed with TransClim and EMAC (conducted within VEU1) are compared. The first and second columns show the relative and absolute changes simulated with TransClim. The third column shows the absolute changes simulated with EMAC (taken from fig. 6 in Hendricks et al. (2018)). The white line indicates the tropopause.

below 500 hPa. In VEU1, a different emission inventory is used than for TransClim. As the OH chemistry is very sensitive to emissions, this can lead to different OH mixing ratios in VEU1 than the ones obtained from TransClim.





The results of VEU1 simulations in fig. 9 are averaged over three years. In contrast, TransClim shows a one-year-average. The good agreement between TransClim and VEU1 shows that the LUTs consisting of one-year simulations are sufficiently good to describe the $NO_x$, $O_3$ and OH change derived from a three-year EMAC simulation.

TransClim also enables to determine the $O_3$ impact of only German road traffic emissions on climate without the requirement of scaling emissions to enhance the signal-to-noise ratio (see also Hendricks et al., 2018). An additional simulation with TransClim is performed in which all road traffic emissions in Germany are neglected. To obtain the climate impact of German road traffic emissions, the TransClim simulation without German road traffic emissions is subtracted from the reference simulation ("reference simulation" - "no German road traffic emissions simulation"). The resulting $O_3$ and OH changes are

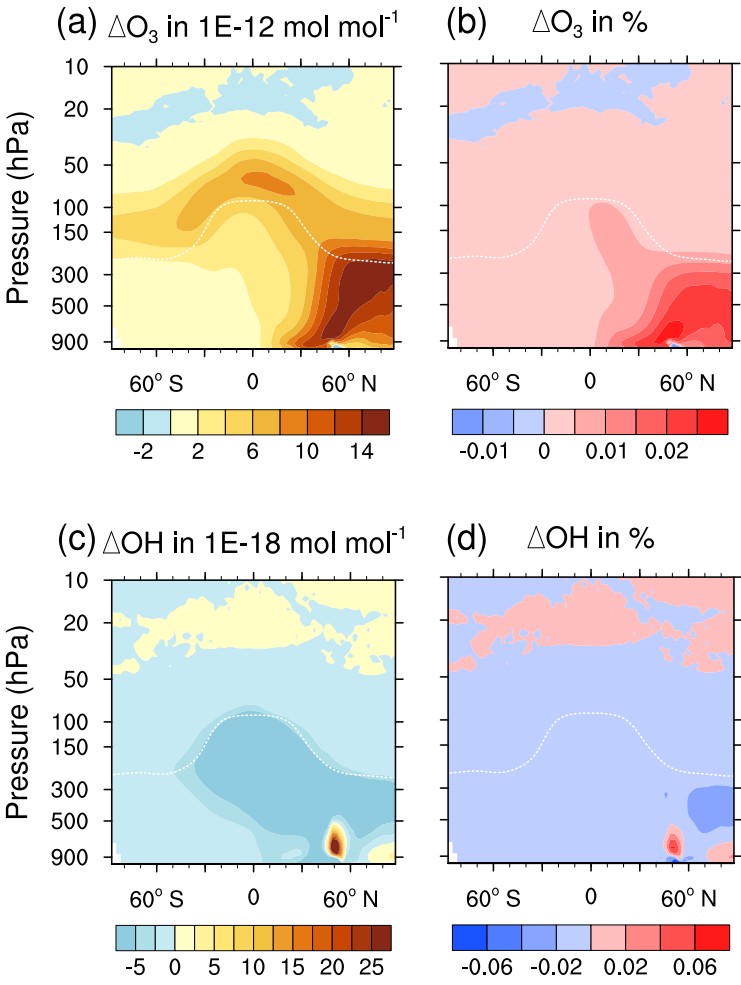

**Figure 10.** Zonal mean of relative and absolute $NO_x$, $O_3$ and OH change caused by German road traffic emissions for the year 2030. The simulation is performed with TransClim. The white line indicates the tropopause.





shown in fig. 10. The pattern of the $O_3$ increase is very similar to $O_3$ change caused by the European road traffic emissions (fig. 9). But the magnitude of $O_3$ change is smaller for German as for European road traffic emissions as the amount of road traffic emissions released by Germany is smaller. The zonal mean of $O_3$ rises by up to 0.03 % in the lower troposphere of the Northern Hemisphere. Noteworthy, a small $O_3$ decrease is observed in the lowermost atmospheric layers at 50°N. In this region, $NO_x$ is strongly increased by German road traffic emissions. The $O_3$ decrease due to a $NO_x$ increase indicates that this

region is "VOC-limited". Considering OH, a decrease due to German road traffic is found in the free troposphere. However, a small increase of up to 0.06 % is observed in the lowermost atmospheric layers at 50°N.

| Variable | Model | Europe | Germany |
|---|---|---|---|
| RF($O_3$) in mW m$^{-2}$ | VEU1 | 1.29 | 0.072 |
| | TransClim | 1.34 | 0.089 |
| $\tau_{CH4}$ change in % | VEU1 | -0.084 | -0.0047 |
| | TransClim | -0.018 | 0.00089 |

**Table 4.** Ozone radiative forcing (RF($O_3$)) and $CH_4$ lifetime ($\tau_{CH4}$) change for the simulation derived in VEU1 (Hendricks et al., 2018) and computed by TransClim for the year 2030. The column "Europe" shows the results for the European road traffic emissions, the column "Germany" describes the values for the German road traffic emissions.

The $O_3$ radiative forcings and the change in $CH_4$ lifetime for the year 2030 are derived from the TransClim simulation and compared with the VEU1 results in table 4. The $O_3$ radiative forcing caused by European road traffic emissions is 1.34 mW m$^{-2}$ and deviates by only 4 % from the VEU1 value. The $O_3$ forcing for the German road traffic emissions is 0.089 mW m$^{-2}$ (derived

with TransClim). It differs from the value obtained in VEU1 by 24 %. This is not surprising as in VEU1 the German values are obtained by downscaling the forcing from the European road traffic emissions (see above). For the change in $CH_4$ lifetime caused by European road traffic emissions, TransClim obtains a significantly lower value than VEU1. On the one hand, the OH increase obtained by TransClim is smaller than in VEU1 (compare to fig. 9). On the other hand, the $CH_4$ lifetimes of the simulations for TransClim's LUTs (about 7.7 years) are generally lower than of the EMAC simulations used for VEU1 (about

8.5 years). This can be caused by the different emission inventories used for TransClim and VEU1 simulations. Moreover, different methods for calculating the $CH_4$ lifetime can cause different $CH_4$ lifetimes and thus influence variations in $CH_4$ lifetimes (Lawrence et al., 2001). Interestingly, the $CH_4$ lifetime change due to European road traffic emissions is negative. But for German road traffic emissions, TransClim computes a positive lifetime change. Due to the downscaling in VEU1, a change in sign can not be reproduced. The change in sign of the $CH_4$ lifetime in TransClim results is caused as for the European

road traffic emissions tropospheric OH increases by 0.03 %. However, for the German road traffic emissions, tropospheric OH decreases by 0.003 %.

To estimate the $O_3$ radiative forcing for different years in VEU1, Hendricks et al. (2018) scaled the $O_3$ radiative forcing with the $NO_x$ emissions from road traffic. Using the emission scaling factors of table 3, TransClim also computes the $O_3$ radiative forcings for these years. Table 5 presents the $O_3$ radiative forcing estimated from the VEU1 simulations and from TransClim.





| Variable | Model | 2008 | 2020 | 2030 |
|----------|-------|------|------|------|
| RF(O$_3$) | VEU1 | 0.28 | 0.13 | 0.07 |
| | TransClim | 0.25 | 0.11 | 0.09 |
| RF(O$_3^{tra}$) | TransClim | 0.44 | 0.22 | 0.15 |

**Table 5.** Radiative forcing of ozone change (O$_3$) and contribution change (O$_3^{tra}$) in mW m$^{-2}$ due to German road traffic emissions for the years 2008, 2020, 2030. The results for the VEU1 simulations with EMAC (Hendricks et al., 2018) and TransClim are given.

The O$_3$ radiative forcing obtained by VEU1 decreases in future. This decreasing trend is well reproduced by TransClim. However, the values differ by 0.02 mW m$^{-2}$. TransClim obtains lower forcings for 2008 and 2020 and a larger forcing for 2030. The radiative forcing of O$_3^{tra}$ from German road traffic emissions obtained by TransClim is also given in table 5. It is about twice as large as the radiative forcing due to total O$_3$. This indicates that the effect of German road traffic emissions on the radiative forcing is underestimated by a factor of two when only the total O$_3$ mixing ratios and not the O$_3$ contributions are 360 regarded (in agreement with Mertens et al. (2018)).

Summing up, TransClim reproduces the results obtained by EMAC very well. Although TransClim underestimates the results of EMAC slightly, it performs very well when being directly compared to EMAC (deviations are below 7 %). It also reproduces the simulation performed in VEU1 satisfactorily well. Moreover, the overall pattern of European road traffic emissions is described very well by TransClim. Only OH is smaller leading to a lower CH$_4$ lifetime change.

## 365 6 Assessment of TransClim

As shown above, TransClim efficiently determines the O$_3$ effect of road traffic emission scenarios on climate. The algorithm used in TransClim (see sect. 2.3) reproduces the results obtained with the global chemistry-climate model EMAC very well.

TransClim considers the emission species NO$_x$, VOC and CO and computes the mixing ratios of O$_3$ and O$_3^{tra}$ in the atmosphere. Thus, the algorithm fulfils requirement (1) of section 2.2. By interpolating within the LUTs, the non-linearity of 370 tropospheric O$_3$ chemistry is regarded (requirement 2). Furthermore, the road traffic emissions are split up into eleven emission regions. For each emission region, own LUTs are set up. Hence, the effect of different emissions regions is included in the algorithm (requirement 3). As TransClim sets up a LUT for each grid box of an EMAC simulation, it can determine the pattern of a variable change. Consequently, TransClim calculates not only the global and tropospheric means, but also the regional effect caused by an emission scenario (requirement 4). Moreover, the method is not only applicable for the determination of 375 O$_3$ and O$_3^{tra}$, but also for other variables such as OH and OH$^{tra}$ as well as the radiative forcings of O$_3$ and road traffic O$_3^{tra}$ (requirement 5).

However, the consideration of O$_3$ background levels is not given with the current approach (requirement 6). The emission variations are bound to a specific base year with a certain O$_3$ background. Varying the road traffic emissions for this base year





results in specific $O_3$ changes. Since the tropospheric $O_3$ chemistry is strongly non-linear, varying the road traffic emissions for
different $O_3$ backgrounds may result in a completely different $O_3$ change. The influence of the $O_3$ background concentration
will be regarded in future studies.

The algorithm used in TransClim determines the climate effect of an emission scenario efficiently (requirement 7). For
example, to compute the global mean climate effect of an emission scenario in one emission region, TransClim needs 0.2 s.
Calculating a three-dimensional variable for one emission region, it takes up to 15 min on a standard computer. For the
determination of the total concentrations such as $O_3$, OH and $NO_x$, the algorithm obtains very good results: the computed
values deviate only little from the values obtained by EMAC (less than $10^{-3}$ %, see fig. B1). The results of the contributions
of road traffic emissions such as $O_3^{tra}$, $OH^{tra}$ and $NO_x^{tra}$ deviate larger (less than 7 %). But the deviations are still so small that
they do not restrict the application of TransClim.

Overall, TransClim fulfils almost all requirements of sect. 2.2 and thus performs very well.

**7  Summary and conclusions**

The response model TransClim efficiently quantifies the $O_3$ effect of road traffic emission scenarios on climate. Considering
the road traffic emission species $NO_x$, VOC and CO, TransClim computes the change in atmospheric variables such as $O_3$, OH
and $NO_x$ as well as the stratospheric-adjusted radiative forcing of $O_3$. TransClim bases on lookup-tables which contain pre-
calculated relations of emissions and their climate effect. These relations are simulated by the global chemistry-climate model
EMAC. Road traffic emissions are divided into eleven emission regions (Germany, Western Europe, Northern Europe, Eastern
Europe, Southern Europe, North America, South America, China, India, Southeast Asia and Japan/South Korea). TransClim is
able to consider emission scenarios in which road traffic emissions of $NO_x$, VOC and CO are varied from 0 to 200 % in each
emission region.

The algorithm used in TransClim is able to compute the climate effect of road traffic emission scenarios very fast. Running
on a standard computer, TransClim is ca. 6000x faster than the global chemistry-climate model EMAC running on a high-
performance computer. For example, it takes 0.2 s to calculate the global mean climate response of one emission scenario. In
other words, TransClim needs approximately $4.5 \cdot 10^5$ less computing time than a climate simulation with EMAC. Hence, it
offers a suitable tool for assessing a broad range of road traffic emission scenarios. As TransClim further considers the tagging
method, it allows for calculating not only the changes in atmospheric composition but also the contribution of road traffic
emissions.

The comparison of TransClim simulations with EMAC simulations (which have not been used for the training to set up
TransClim) shows that TransClim is able to reproduce the changes in chemical species and in radiative fluxes very well. The
comparison of TransClim with an equivalent EMAC simulation reveals that the errors are small ($0.01 - 7$ %) and thus do not
hamper the application of TransClim.

However, the current setup of TransClim restricts its range of usage. The LUTs are generated from emission variation
simulations with the global model EMAC. This enables to compute the global and regional atmospheric response. But to





calculate the response on a local scale, it is mandatory to perform additional simulations with models such as the climate model MECO(n) (coupled model system MESSyfied ECHAM and COSMO models nested n-times; Kerkweg and Jöckel, 2012a, b) which can have a finer grid resolution ($0.44°$). Furthermore, the LUTs base on emission variation simulations of the year 2010.

For a different time period, the concentration of the $O_3$ background may vary significantly and hence, the current LUTs would not be valid any more. Consequently, new LUTs need to be created considering the climate response of a very different $O_3$ background concentration. Moreover, the current LUTs consider only variations of road traffic emissions. To include the $O_3$ response of other land based traffic modes on climate such as railways and shipping, additional emission variation simulations are required to generate new LUTs.

Overall, the approach used for TransClim is very flexible. It enables to easily extend the LUTs with additional emission regions, traffic modes and years. However, the computational resources required for emission variation simulations is high and hampers the extension of the LUTs. But once the LUTs are generated, TransClim is able to quickly compute the $O_3$ effect of an emission scenario on climate.

The impact of traffic emissions on air quality and climate is also examined by other response models. For example, the re-
sponse models LinClim and AirClim analyse the climate response of aviation emissions (Lim et al., 2007; Grewe and Stenke, 2008; Grewe et al., 2012; Dahlmann et al., 2016). Both models use a linear approach to compute the $O_3$ change in the strato-sphere. In comparison to the lower troposphere, the $O_3$ chemistry in the upper troposphere and stratosphere is not dominated by strong non-linearities. Thus, the linear approach for determining the $O_3$ concentration in the stratosphere works well for LinClim and AirClim. However, these approaches would not work for TransClim as the road traffic emissions are released into
the lower troposphere where the non-linearities of the $O_3$ chemistry are an important factor to be considered.

Another example is the model TM5-FASST. It investigates the impact of pollutants such as $NO_x$, $SO_2$, CO and BC on air quality (Leitão et al., 2013). Moreover, TM5-FASST calculates radiative forcings, temperature variations, mortality and the impact on vegetation and crop yield. But this response model also uses a linear approach for computing the $O_3$ change. Thus, large deviations (of up to 20 percent points) are found in regions with high emissions of $O_3$ precursors. Furthermore, TM5-
FASST considers the influence of the precursors $NO_x$, VOC and CO on the $O_3$ chemistry separately. As TransClim interpolates within the LUTs which base on $NO_x$, VOC and CO emissions simultaneously, it considers the influence of the three precursors in producing $O_3$ all together. Moreover, it regards the non-linearity of the tropospheric $O_3$ chemistry. Consequently, it produces by far less deviations than TM5-FASST. Even though, TM5-FASST determines more impact metrics, it does not regard the contribution of $O_3$ precursors. Thus so far, no other response model than TransClim analyses the climate impact as well as the
contribution of road traffic emissions. This makes TransClim a unique model.

Summing up, TransClim is able to quantify the climate effect of $O_3$ changes caused by road traffic emission scenarios reliably. However, further developments are planned. To assess the climate effect of future emission scenarios, the impact of different $O_3$ background concentrations needs to be included in TransClim. Moreover, the radiative forcing caused by a change of methane lifetime will be regarded in TransClim as well. Besides, the integration of other traffic modes such as
shipping is desirable to expand the applicability of TransClim. The current implementation regards only the climate metric stratospheric-adjusted radiative forcing. To provide deeper insight into the climate effect, further climate metrics such as surface



temperature change need to be integrated. In addition, road traffic emissions also affect aerosols. The inclusion of the aerosol effect in TransClim would complete the assessment of mitigation strategies. Despite these planned extensions of TransClim, the response model is operational and ready to assess the $O_3$ effect of mitigation options for road traffic on climate.

*Code and data availability.* The exact version of the model TransClim as well as the corresponding EMAC simulations for the lookup-tables used to produce the results presented in this paper are archived at the German Climate Computing Center (citation).

**Appendix A: Tagging method**

To attribute the effect of road traffic emissions to tropospheric ozone, we use a *tagging method* (Grewe et al., 2010; Grewe, 2013; Grewe et al., 2017; Rieger et al., 2018). It considers ten source categories: emissions from the sectors anthropogenic
non-traffic (e.g. industry and households), road traffic, ship traffic, air traffic, biogenic sources, biomass burning, lightning, methane ($CH_4$) and nitrous oxide ($N_2O$) decompositions and stratospheric ozone production. The tagging method computes the contributions of these ten source categories to seven chemical species or chemical families: $O_3$, hydroxyl radical (OH), hydroperoxyl radical ($HO_2$), CO, peroxyacyl nitrates (PAN), reactive nitrogen compounds ($NO_y$, e.g. NO, $NO_2$, $HNO_4$, ...) and non-methane hydrocarbons (NMHC). Like an accounting system, this method follows all important reaction pathways for
the production and destruction of the regarded species.

As an example, a bimolecular reaction of the chemical species A and B forming the species C is considered (see also Grewe et al., 2010):

$$A + B \longrightarrow C \tag{A1}$$

Each species $A, B$ and $C$ is split up into the ten subspecies $A^i, B^i$ and $C^i$. Thus, $A^i$ describes the *contribution* of the source
category $i$ to the concentration of $A$ (the same holds for $B^i$ and $C^i$). These tagged species ($A^i, B^i, C^i$) go through the same reactions as their main species ($A, B, C$). In general, if A from the category $i$ reacts with B from category $j$, the formed $C$ is counted half to the category $i$ and half to the category $j$:

$$A^i + B^j \longrightarrow \frac{1}{2}C^i + \frac{1}{2}C^j \tag{A2}$$

Regarding all possible combinations of the reaction of $A^i$ with $B^j$, the production of $C^i$ is deduced mathematically by a
combinatorical approach and eventually leads to (see Grewe et al. (2010) for more details):

$$ProdC^i = \frac{1}{2}kAB\left(\frac{A^i}{A} + \frac{B^i}{B}\right) \tag{A3}$$

with $k$ being the reaction rate coefficient of reaction A1. Consequently, this combinatorial approach enables a full partitioning of the reaction rate. In this manner, the tagging method used here determines the contribution of road traffic emissions to ozone ($O_3^{tra}$).





**Appendix B:  Evaluation of TransClim: Relative frequency distributions**

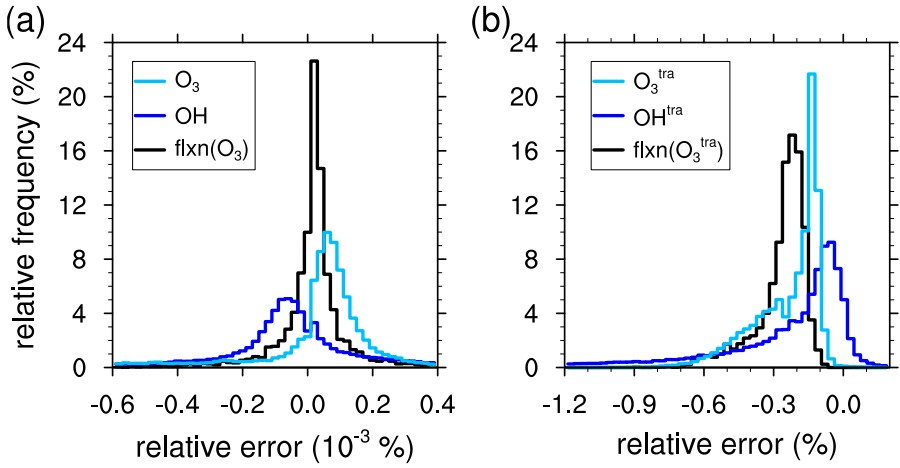

**Figure B1.** Frequency distributions of the relative differences between the simulations performed with TransClim and EMAC. The distributions are shown for the variables $O_3$, OH and the radiative fluxes caused by $O_3$ (flxn($O_3$)) as well as the corresponding contributions. The plot includes only the values for all grid boxes in the troposphere of the Northern Hemisphere. For flxn($O_3$) and flxn($O_3^{tra}$), only the values at top of the atmosphere are taken into account.

The relative frequency distributions of the relative differences between the TransClim and EMAC simulation are shown in fig. B1. The simulation setup is described in sect. 5.1. The distributions base on all grid boxes in the tropospheric Northern Hemisphere. Fig. B1 shows the relative frequency distributions for the variables ozone ($O_3$), hydroxyl radical (OH) and net flux at top of the atmosphere (flxn($O_3$)) as well as the corresponding contributions ($O_3^{tra}$, $OH^{tra}$, flxn($O_3^{tra}$)). For $O_3$, OH and
flxn($O_3$), the relative errors are very low. Most of the grid boxes do not exceed errors larger than $0.5 \cdot 10^{-3}\%$. The relative errors of the contributions are significant larger. Few grid boxes of $OH^{tra}$ deviate by more than 1.2 %. Regions with larger deviations occur in the upper troposphere of the Southern Hemisphere (see also fig. 7) where the contributions of $OH^{tra}$ are generally low. Dividing the absolute differences by these small values leads to large relative differences.

*Author contributions.* Vanessa Rieger designed the model concept, implemented the model, performed the simulations and evaluations and
wrote the paper. Volker Grewe conceived the model concept, coordinated its development and significantly contributed to the interpretation of the results and to the text.

*Competing interests.* The authors declare that they have no conflict of interest.



*Acknowledgements.* This study was supported by the DLR transport program (project "Transport and the Environment – VEU2"). The EMAC simulations were performed at the German Climate Computing Center (DKRZ, Hamburg, Germany), which also provided kind sup-
port for long-term storage of the model output analyzed in this work. We used the NCAR Command Language (NCL) for data analysis and to create the figures of this study. NCL is developed by UCAR/NCAR/CISL/TDD and available on-line at http://dx.doi.org/10.5065/D6WD3XH5. We thank Axel Lauer from DLR for very helpful comments which improved the article.



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
