# Peer review of "TransClim (v1.0): A chemistry-climate response model for assessing the effect of mitigation strategies for road traffic on ozone"

_Geoscientific Model Development, 2021_

## Author Comment (AC2)

**Response to Reviewer #1**

*We would like to thank the reviewer for the helpful comments on the manuscript. It helped to improve our manuscript. Please find our comments in italic.*

This publication describes the development of a parameterized source-receptor model at global scale to have a tool for fast assessment of the atmospheric impacts of (changes in) road transport emissions. The paper makes the impression that it was written a few years ago, as some relevant recent references were not included. Overall, it was not clear to me, why this model would be specific to road-transport emissions, as the modeling principle could be relevant for other source types as well. While the ability to describe source contribution and source emission sensitivity was one of the 'sales' arguments for the paper, the element of describing how the 'contribution' element is tested,and how it is used for assessment was not well developed in the paper. The paper is perhaps also a bit too much overselling it's uniqueness. While it is good to have several modelling tools that can make rapid impact assessments, there are now several assessment tools in the literature, including the TM5-FAST model, but also e.g. Wild et al. 2012, and Butler et al. (2018) https://gmd.copernicus.org/articles/11/2825/2018/

> *Thank you for your comments. As also suggested by reviewer 2, we thoroughly restructured the method section. The description of the tagging method and the introduction of the term contribution are now at the beginning of the method section. This makes the usefulness of the terms contribution and impact clearer to reader. To our knowledge, so far no other response model assesses the contribution (determined by a tagging method) and the impact of road traffic emissions together.*
> *As suggested, we added the references to our manuscript. For example, the study Wild et al. (2012) is compared to TransClim in the conclusions. The study Butler et al. (2018) describes another tool for source attribution of tropospheric ozone. As we use the tagging scheme introduced by Grewe et al. (2010), the paper of Butler et al. (2018) does not directly relate to the response model TransClim. Thus, we did not include it in the assessment of the response models in the conclusion. Furthermore, Van Dingenen et al. (2018) provides indeed a similar tool as TransClim, however, with a different approach and with different impact metrics. As discussed in the conclusion, the approach used in TransClim is not necessarily bound to road traffic emissions. It can be easily extended on other traffic modes or other emission sources. For now, the goal of TransClim is to assess the climate effect of mitigation strategies for road traffic.*

I recommend to publish the paper with major revisions, after considering the comments above and the detailed remarks below.

Detailed remarks:

and particulate matter- along with NO2 one of the most important pollutants- a bit strange to leave it out in this list, even your study is not focussing on it.
> *As suggested, we included "particulate matter" in the list.*

it may be worth to mention here the characteristic of the model that make it specific for transport. In general such models do not need to be specific for transport, but could be applied to any sector.
> *The approach of TransClim is indeed very flexible. It is easy to extend the LUTs to include other transport modes or even other emission sectors. This is discussed in the conclusion.*

I understand that you define impact on two dimensions: response and contribution. This needs to be clearer highlighted in the abstract. The combined approach to contribution and responses (I think) is what makes Transclim and this analysis special. (?)

> *We added more explanation to the abstract.*

and probably more than this, including methane (for certain engine types) and HCFCs/HFCs from airconditioers.

> *Yes, this is an incomplete list of the pollutants emitted by road traffic. We adapted the sentence.*

here you correctly include PM (but not in abstract). And PM is influencing ozone through heterogeneous reactions.

> *Thank you, we added particulate matter in the abstract.*

Most studies? Are there studies that don't and why?

> *In a few cases, road traffic emissions can also decrease winter-time ozone in the troposphere. We added an example to the text.*

What did Reis and Tagaris find? Sentence is is now without purpose.

> *Thank you, we gave more information on the study Tagaris et al. (2015).*

Explain better what the 0.8 % refers to. Global ERF; German ERF, 0.8 of world traffic? I can imagine many possibilities…The 5 mK is a global number, or Germany?

> *Hendricks et al. (2018) determines the contribution of German road traffic emissions to the total anthropogenic stratosphere-adjusted radiative forcing and the global mean surface temperature change. We adapted the sentences accordingly.*

to 52. This paragraph, which is essentially the overall method is not terribly well elaborated. I propose you first indicate that you want a method and tool that can both assess sensitivity to emission changes, and current contribution and call that impact. I don't really see why an emission perturbation is solely defining impact. Later you define 'total effect' an unnecessary and ambiguous term. Clear language may help the understanding. L. 43- I only partly agree that perturbation method do not take non-linear relations into account- for instance a perturbation on a preindustrial situation would give a very different result from the perturbation of present-day. I propose: "depending on the chemical regime, large errors may occur when extrapolating emission perturbation relationships to larger perturbations." L44 'it quantifies only impact of emissions': apart from what is impact, I think it is important that you talk here about emission perturbations (sensitivities).

> *This paragraph gives an overview of the methods commonly used and introduces the perturbation and tagging method. Thank you for pointing at the necessity to clearly define the terms used. Very much appreciated! We rephrased the paragraph to better explain the terms impact, contribution and effect.*
> *Correct, the perturbation approach is a linearization of the current chemical state. And we agree that for larger perturbations errors may occur. However, this is not the point we made. In non-linear systems the perturbation/sensitivity approach and contribution method must give different results, which are significantly even for small perturbations (e.g. Grewe et al. 2010).*

unclear until this point what you do with methane- from a variety of perspectives. Although direct emissions may not be terribly high, indirect emissions may be more significant, e.g. from oil and gas production. A large part of the fuel is used for transport. Changes in emissions of NOx, VOCs, and CO will affect the lifetime of CH4 on timescales of up to ca. 20 years. There are methods how to include the effect of lifetime changes on CH4 itself (an important effect) and O3. Here or before it should be already be mentioned if/how this is included.

*TransClim computes the contribution of various emissions and precursors to the methane destruction and by that the contribution to methane lifetime as well as changes caused by OH changes. Methane as an ozone precursor is not yet regarded. Direct emissions of methane from road traffic are small and thus neglected. We added this information to the introduction.*

The graph is not extremely informative, as it doesn't provide much insight in 1) the scale of emission perturbation (grid, region, world?), the type of perturbation (annual, monthly, all components together, or separate, size of perturbation), time scale of effects, equilibrium or transient…. Is this figure needed?

*We deleted the figure and rephrased the paragraph. It gives an overview of how TransClim works and thus guides the reader through the method section.*

This section is in part not requirements but rather a description of assumptions. Rename?

*We have formulated requirements which are important for the performance of TransClim. We actually tested various methods in Rieger (2018) and the algorithm which performs the best is presented in this manuscript. Here we only refer to the results in terms of conclusions. We have reformulated the text to clarify this.*

where is CH4?

*This study focuses on the effect of road traffic emissions on ozone. The effect on methane is a side aspect which is evaluated for German road traffic emissions in table 5. The requirements are similarly applicable to methane.*

As explained before, most models consider non-linearity. The point is that the nonlinear response to changes should be computed within I a certain margin of accuracy.

*Most response models which are available in this research field do not consider the non-linear behaviour of the tropospheric ozone chemistry (see conclusion). They assume a linear relationship between emission change and ozone change. Consequently, it is important that TransClim regards this non-linearity. As you mentioned, it is further important that the deviations between the results computed by TransClim and EMAC remain low. This condition is covered by requirement (6). Please see also our comment above.*

Explain why you think the choice of these big continental scale regions is appropriate for the problems that need to be quantified.

*The emission regions are chosen in such a way that the climate effect of road traffic emissions from different part of the world can be evaluated. The source-receptor relations differ for different regions. This set of emission regions is not fix. If there is a desire to refine the emission regions over a particular region, then new emission variation simulations need to be performed with EMAC and integrated to the LUTs of TransClim. On purpose, we build up the algorithm of TransClim very flexible, so the integration of new emission regions is very easy (see also section 2.4.2 and conclusion). We added further explanation to section 2.4.2. Besides, there is a large difference in local air quality responses and climate impact responses. These emission regions are not applicable for air quality assessments.*

l03 Again here: calculate is one thing, but with which accuracy. Is this ERF or RF. In either case to what extent is this state of the art and method?

*We consider the stratosphere-adjusted radiative forcing here. The stratosphere-adjusted radiative fluxes are calculated by the global chemistry-climate model EMAC which is a state-of-the-art model. The question how accurate can the algorithm of TransClim determine the stratosphere-adjusted radiative forcing is evaluated later in the manuscript in section 3 "Model evaluation".*

Background refers to a hypothetical situation without (transport?) emissions. Is this what is meant. Or do you rather mean that the large ozone trajectory according to socioeconomic and technological assumptiosn as used in the climate community should be considered. Also note that the RCps are now superseded by SSPs (with some consequences for emission trajectories). This is not a major issue for the concept, but this could be mentioned somewhere.

> *As suggested by reviewer 2, we omitted this point here to focus on the points which TransClim is able to do. We added the consideration of the background ozone concentrations as future improvements of TransClim in the conclusion section.*

this is about specifying efficiency and accuracy.

> *Yes, this is correct. But the response model has to be efficient and provide low errors; otherwise it does not provide an additional benefit. Thus, efficiency and accuracy is a requirement for the algorithm.*

(Figure 2). Please clarify if the red dots are representing what has be done in terms of perturbations. In this case it may be a bit problematic that not more perturbation lower than 1 have been implemented, as in several world regions this may be the overall trajectory that emissions are going already, and will even more so go in future. It is also not very clear how the point 6 is taken into account (changing baseline ozone).

> *The red dots are just a sketch of the emission scaling and do not represent the actual emission variation simulations. These can be found in table 2 now. We indicated this in the text. As mentioned in the conclusion, the requirement 6 could not be regarded by this algorithm and will be investigated in future studies.*

what is a standard computer? What do you calculate for an emission scenario, each year, every 10 years?

> *Here, a standard computer describes a work station, in contrast to a high performance computing system. We noted this in the text as well. The runtimes are given for an arbitrary emission scenario in one emission region. The runtimes are not depended on the emission scenario, but on the number of performed interpolations which depend on the number of considered emission regions.*

138-146 in an earlier part of the text it should already be explained what problems need to be solved, and why these large regions are appropriate for this.

> *The current set of emission regions is not fix. If needed, additional emission variation simulations can be performed with EMAC for smaller emission regions. We added application examples for TransClim in the introduction.*

156-169 Again some further rationale for this model set-up should be provided. Nowadays (2021) 2.8x2.8 doesn't look very state-of-the art (e.g. look at the Van Dingenen paper (2018), that use a 1x1 resolution. Is the high vertical resolution needed in view of the course horizontal resolution? Why 'free running' (I assume you mean not constrained by (re-) analysed data?)- where there could be clear advantages of putting some constraints- e.g using prescribed SST or nudging. Is the explanation in l. 176?

> *In terms of resolution, the model we used has a simulation resolution compared to CMIP models from ACCMIP (Lamarque et al. Geosci. Model Dev., 6, 179–206, 2013, www.geosci-model-dev.net/6/179/2013/) as well as for the new runs in ACCMIP, which are currently published. EC-Earth3-AerChem (van Noje et al. 2021, https://gmd.copernicus.org/articles/14/5637/2021/) has a 2° by 3° resolution for chemical tracer transport.*
> *Yes, free running is referring to a climate simulation, where the atmospheric circulation is calculated by the primitive equations and not prescribed by atmospheric winds etc. This is now explained in the text.*

I have no idea what a QCTM mode is- abbreviations need to be duly explained
> *We repeated the explanation of the abbreviation here again and further referred to the paragraph above where this abbreviation was explained.*

173-184. Overall this makes a sound impression, it would help the reader to explain why this is important, and what kind of 'improvements' are found compared to more conventional 'off-line' calculation of radiative forcing.
> *We reformulated the whole paragraph.*

MaccCity (if I remember well based on EDGAR3) is pretty old by now- and goes up to2000 (?). I understand that the development of this paper has been taken a while, but there are now inventories like EDGAR5, CEDs that have updated emissions with more recent years.
> *MACCity provides emission data for anthropogenic and biomass burning for the period 1990 - 2010. It is based on the data set ACCMIP and RCP8.5. In this study, we provide a data set for the LUTs for the year 2010. However, the approach used in TransClim is very flexible. So it is easy to extend the LUT with more emission variation simulations of more recent years.*

see similar information in Van Dingenen.
> *Thank you for this hint. We added the reference to the manuscript.*

Can you specific how many simulations are available, and also how the change baseline according to RCP was considered? As a first dimension?
> *For each emission region, 21 emission variation simulations have been performed. We added this information and an additional table (table 2) to the manuscript. The current setup of TransClim is not able to consider changes of ozone background concentrations. This will be regarded in future studies, as discussed in the conclusions.*

This synthetic emission is instructive, intuitively I would say that one can still expect problem in Northern Europe (as well as western Europe, Germany) where 'titration' effects can mess up the analysis.
> *As the evaluation of this test case shows, TransClim reproduces the results obtained by EMAC very well. In the source region Europe, the errors remain below 0.3%. In particular, over Northern Europe, TransClim underestimates the EMAC results at most. But still, the errors are so small that they do not hamper the applicability of TransClim. While titration is an important aspect for air quality assessments, we focus on the climate impact here.*

What results are you talking about here? SARF globally for one year?
> *As suggested by reviewer 2, we deleted figure 6. It showed the tropospheric $O_3$ and $O_3^{tra}$ columns in Dobson units obtained by TransClim and the relative errors towards EMAC. Now, Figure 5 shows the relative errors between TransClim and EMAC results and figure A1 in the appendix show the absolute values of the tropospheric $O_3$ column, the tropospheric mean of OH and radiative flux of $O_3$ at top of the atmosphere as well as the corresponding contributions.*

are you really discussing O3 or the O3 RF?
> *The former text was related to $O_3$ and its contributions of road traffic emission to ozone $O_3^{tra}$. We rewrote the whole paragraph.*

underestimation of what? I am not sure that 7 % deviation is 'very low', this could link to the specification section earlier
> *We rephrased the whole section.*

Clarify whether these are transient simulations, equilibrium or something else?

*VEU1 simulations are equilibrium chemistry simulations for 2030 and the climate response is calculated transient based on the transient changes in emissions and the results of the equilibrium simulations.*

I think it is a bit confusing to the readers to call it a German emission scenario and consequently apply it to all of Europe. Can the authors expand on the 'robust' signal issue? Is this an apparent drawback of having a model unconstrained by analysed meteorology?

*Although the noise of the chemical perturbation is significantly reduced by the QCTM mode of EMAC, it may be still challenging to quantify the climate effect of a small signal resulting from a change in German road traffic emissions by the perturbation method. To avoid this problem, Hendricks et al. (2018) enhanced the signal and perturbed the European transport emissions. The climate response of German transport emissions is determined by downscaling the European response with the ratio of German to European $NO_x$ emissions. Hendricks et al. (2018) assess this scaling procedure as follows: "Estimating the German effect by this scaling procedure requires the assumption that the radiative forcing per emitted amount of pollutant is similar for the European and the German emissions. Since pollutants released over Europe usually experience vigorous mixing, uncertainties due to this assumption are probably small." We adapted the text.*

In view of the previous remarks: perhaps for this paper it is not very necessary to highlight the Germany case- it sounds a bit like a 'patch' to me.

*Agreed, the procedure might be seen as a 'patch'. The method that we apply here (TransClim) reduces the signal-to-noise ratio. And hence, TransClim is capable of addressing the effects of German emissions, only. Although the global impact is minor, it is still of interest to compare different climate mitigation options. Therefore, we would like to keep this passage. We refer to the differences between European and German emission effects in e.g. table 5. In addition this information is necessary to understand the reply wrt. the referee's comment for line 393.*

Finally CH4. But what is done with this information?

*German road traffic emissions influence not only the tropospheric ozone but also the lifetime of methane. Both are important greenhouse gases. We added this intention to the text.*

3 years. Clarify if you mean 3 years from a transient simulation, or what?

*VEU1 Simulations are equilibrium chemistry simulations for the year 2030. The resulting climate response is determined transient (see in detail comment above).*

'only' is normative language. 24 % seems high. How is it comparing gto the specs?

*TransClim reproduces the ozone radiative forcing of European road traffic obtained by EMAC very well. Here, the deviation is only 4 %. But for German road traffic, the deviation is significantly larger (24%). However, this can be taken as a benchmark, because this deviation is caused by the different methods how the climate effect of German road traffic emissions is determined: TransClim determines the climate effect by comparing a simulation with German road traffic emissions to a simulation without German road traffic emissions. In VEU1, the climate effect of German road traffic emissions is determined by downscaling the European climate effect (see also above).*

346-350 Indeed interesting, but unfortunately without explanation.

*We added a possible explanation.*

standard computer?

*A standard computer describes a work station. This information is already added in section 2.5.*

Probably a more authorative publication is Van Dingenen et al. (2018), which also extensively describes methodology, error analysis against a range of issues (deviation from linearity, deviation from 'additionality', using a wide range of high/and low end scenarios, and comparison with other literature estimates of similar scenarios. Although the Van Dingenen paper does not give a detailed regional analysis of ozone columns and RF, the analysis shows e.g. for 2030 deviations for summer surface ozone in the order of 4- 9 % for most regions under a high emission scenario and 8-13 % for a low scenario (with an outliers of around 20 %). However, this includes effects of CH4, and by far more regions that in the current study. Interestingly an comparison with a range of publications including results of AR5 showed that FASST was well within a range of other scenario results. Based on the analysis in this paper, I can not support the statement that deviations are 'far less' than from FASST- given the much more limited scope of the evaluation.

*Thank you. We added the study of Van Dingenen et al. (2018) to our manuscript. Based on this study and the evaluation given in our manuscript, a direct comparison of the errors is difficult because different metrics are given. As also suggested by reviewer 2, we added more evaluation simulations to our manuscript. These show that TransClim performs well and the deviations from the full global model EMAC remain generally below 10%. We adapted the paragraph.*

TM5-FASST does include ozone precursors (including CH4)- so it is not clear what is meant with this sentence.

*To our knowledge, TM5-FASST does not compute the contributions of a specific emission sector to the ozone concentration using a tagging method. We modified the text accordingly.*

The study by O. Wild et al. (2012) on the HTAP included a parameterisation of non - linearities of ozone. https://doi.org/10.5194/acp-12-2037-2012, 2012.

*Thank you for pointing out this study. We added a paragraph about it to the conclusions.*

---

## Author Comment (AC3)

**Response to Reviewer #2**

*We would like to thank the reviewer for thoroughly examining the manuscript. We gratefully incorporated the reviewer's comments which certainly improved our manuscript. Please find our replies in italic.*

**Manuscript Summary**

The manuscript describes the development of parameterised source-receptor model (TransClim) to quickly and efficiently assess the climate impacts from changes in road traffic emissions. TransClim uses look up tables that have been built from simulations of the chemistry climate model EMAC. These look up tables define the relationships between different variables (including $O_3$) and the change in NOx, CO and VOC emissions from road traffic sources. This is then linked to radiative forcing to determine the climate impact from road traffic emissions. The specific contribution from road traffic emissions is further quantified by using a tagging method. The paper describes the development of TransClim and presents results of the evaluation against additional EMAC simulations.

I think the manuscript describes a useful new tool to assess the impact of road traffic emissions and I think it should be published once the comments below have been addressed. One of the main issues to be addressed is improvements to the clarity of the methods section. This will allow for a better flow to manuscript and make clearer how TransClim was developed.

**Major Comments**

- I know the focus of this tool is the climate impact of $O_3$ but one of the most important impacts of road traffic is on local air quality and health. This seems to be an equally if not bigger motivator for understanding the impact from road traffic emissions but is not mentioned in the introduction? Also would this method also be applicable to calculating the impacts on surface $O_3$ and therefore an estimate of both the impact on climate and air quality could be obtained?

  *We added an extra paragraph in the introduction describing the impact of road traffic emissions on human health and environment. The method used for TransClim is also applicable to determine the impact on surface ozone and air quality. However, this requires more work, e.g. on particulate matter. We added this in the conclusion.*

- If you are going to mention the methods in the introduction can you provide an improved explanation of what is used in this study (see further comments in minor section below) or just leave the details to the methods.

  *The introduction just gives an overview of the common methods and shortly introduces the tagging method which is used in this study. We improved the explanations given in the introduction. We further moved the description of the tagging method from the appendix to the method section so that the reader gets a clearer comprehension of this method.*

- I think significant improvements could be made to the methods to make the manuscript read better and provide a more logical flow. I found the structure of the methods section confusing as it appeared to be the wrong way round. The first section told us what TransClim does (which already occurred at end of introduction) before how you developed the response model. Also, the development of the look up tables and model simulations that were used to develop these should probably be discussed earlier as these are the key datasets that underpin the tool. Then the methods should discuss how the algorithm was

built from these look up tables. Also Section 2.2 is not a list of requirements and needs to be changed to better reflect what you want to say here. Furthermore, in section 2 there was only a brief mention of the Tagging method but not how it fits in with the rest of the development process. Additionally, the workflow sections and Figure 5 could be improved to make it easier to follow how the tool works (see further minor comments below).

*Thank you, we restructured the method section as suggested. We turned around this section: now we describe the model EMAC and the model simulations for the lookup-tables before the description of the algorithm. We further moved the description of the tagging method and of the interpolation error to the method section. We improved the paragraph and the figure describing the workflow. However, we would like to keep Section 2.1 at the beginning of the section as it gives a good overview of how the model TransClim is working and thus can lead the reader through the method section. We have formulated requirements in Section 2.3 which are important for the performance of TransClim. We modified the text to clarify this (see comment below).*

- The definition of the emission source regions seems slightly arbitrary to me. Why were these eleven regions chosen and is the future use of tool limited by the defined set of regions used to create it? The regions seem heavily weighted towards Europe and particularly broad over other areas like North America and Asia. Can you provide some justification for this choice and any limitations that might occur as a result of applying this tool to regions outside of Europe. Also could the regions be improved by only including land-only points, as currently there is a lot of oceanic parts considered in some regions (Fig. 3).

*The emission regions have been chosen so that the climate effect of road traffic emissions from different part of the world can be considered. However, this set of emission regions is not fix. The emission regions can be easily extended or one emission region can be further refined by performing additional emission variation simulations. Europe and Germany have a finer resolution as the work was embedded in the DLR project "Verkehrsentwicklung und Umwelt 2" which regarded the climate effect of German road traffic emissions. Considering land-points only would not change the amount of emitted road traffic emissions, as no emissions occur over the oceanic parts. We adapted the text in section 2.4.2.*

- The EMAC simulations are based on a single year, 2010, and use emissions for this year from a particular emissions inventory. What are the limitations of basing the tool solely on 2010 emissions, meteorology and background concentrations? How will changes to variables such as these effect the chemical relationships within TransClim and impact the predictions made by the tool?

*The current set-up of TransClim is bound to the year 2010. Slight variations of the background concentrations, emissions and meteorology may not have a large impact on the results. However, for a very different ozone background, the tropospheric ozone chemistry may react very differently on emission changes. For this case, new emission variation simulations for the LUTs are required. This point is discussed in the conclusions.*

- The EMAC emission variation simulations used to build the look up tables that form the basis of the TransClim tool are described in Table S4. These simulations seem disproportionately biased towards the emission source region of Germany and therefore are the results from the TransClim tool from other regions affected by this weighting of training data? Are you therefore assuming that the $O_3$ and emission relationships calculated over Germany can be applied around the world? There are hardly any emission perturbation experiments over Asia or North America for example. This seems quite limiting as I would have potentially

thought they would be quite different emission relationships over India and China to those over Germany. For example, across these regions you are basing the $O_3$ response relationships on only a single perturbation simulation. This seem particularly limiting and depends heavily on your simulations over Germany.

*Unfortunately, the way we set up table S4 was misleading. For each emission region, 21 emission variation simulations have been performed with EMAC. For Germany, 23 emission variation simulations have been performed and this is the reason why it appeared in a different column in table S4. So each emission region is (almost) weighted equally by the set of emission variation simulations. To not confuse the reader, we changed the way of representation: table 2 in the manuscript now shows the 21 emission variation simulations for each emission region. Hence, we do not presume that any relationship found for Germany is applicable to any other region.*

- Whilst I agree that testing the performance of TransClim over Europe is needed, I was disappointed not to see further evaluation across other emission source regions. This would strengthen the evidence of the applicability of the tool in other world regions. I would expect the performance to be better over Europe given the number of EMAC simulations performed over this region to construct the Look Up Tables in Table S4. However, given the lack of EMAC simulations over other source regions e.g. Asia and North America, I would therefore feel it is even more important to evaluate the performance of TransClim for emission perturbations and response over these regions. Given the lack of evaluation over these regions I am not sure how TransClim currently performs on regions other than Europe.

*Good point! As suggested by the reviewer, we tested the performance over different regions in fig. 6 (sect. 3.1.2). We added further evaluation simulations for North America, South America and Asia. Across the different regions, the relative errors of TransClim remain below 10 %. This shows that TransClim is not only applicable over Europe but also over other regions of the world.*

- A description of the linear interpolation error is found in section 5 but probably this should be moved earlier towards the methods section as this is the first point it is mentioned in the manuscript. I also found there was little detailed quantification of this error other than saying the evaluation of a TransClim simulation is similar to EMAC. However, looking at Figure 8 it appears that the linear interpolation error would increase at larger emission perturbations (e.g. 1.0 to 0.0 or 1.0 to 3.0). I think it would be useful to know for what emission scalings does TransClim have low errors and therefore performs best at. Also, it would be good to document for what size of emission perturbations do the relationships used in TransClim breakdown at. This seems a key aspect of confidence in TransClim and should be explored further.

*We moved the description of the interpolation error to the method section. To quantify how TransClim performs for different emission scaling, we added further evaluation in sect. 3.1.2. Fig. 7 shows the relative errors of TransClim for different strength of scaling in one emission region. The errors remain below 4 %. As long as the emission scaling is within the range of the LUT, TransClim performs well.*

**Minor Specific Comments**
Page 1, Line 9 – change "bases" to "is based"
    *We changed the text accordingly.*

Page 1, line 23 – $CO_2$ is not carbon monoxide
    *We changed the text.*

Page 2, lines 24 – 27 – Could also mention the other impacts from road traffic here e.g. the emission of particulate matter and also the importance of O₃ and CH₄ for climate.
*We added this to the text.*

Page 2, line 27 – I think "In general" is quite a broad statement here. Would be good to mention what conditions are important here e.g. time of day, season, urban vs rural etc.
*Thank you for the hint. We further specified it.*

Page 2, lines 28-29 – "not linear", why not just say non-linear?
*Yes, done.*

Page 2, line 36 – any comment on winter changes?
*We added the values for winter changes.*

Page 2, line 37-38 – what does these studies show?
*Thank you, we gave more insights about the results of these studies.*

Page 2, line 38 – remove "by"
*We changed the text accordingly.*

Page 2, line 39 – Is this number global? Also would it be better to just show in K.
*Yes, the surface temperature change is global. We added this information to the manuscript and changed the unit to K.*

Page 2, line 42 – When explaining the perturbation method it is better to refer to the simulations as a control (all emissions) and an experiment (perturbed emissions).
*Thank you, we added these terms to the text.*

Page 2, line 43 – Why does the perturbation method not account for non-linear interactions? I would have thought most chemistry models would do.
*This is an important point, which we obviously did not describe thoroughly enough. All chemistry models have the non-linear chemical relations, independent of the approach used. The statements on linearity apply to the method to calculate the impact and contribution. The perturbation method, by definition, is linearizing the relationship between emission change and ozone response, whereas the contribution method non-linearly analyses reaction pathways and attributes ozone partial concentrations to emissions. Only for linear chemical relationships, the impact and the contribution are the same and the perturbation method is able to determine the contribution of a specific emission sector to, e.g., ozone. However, for non-linear relationships, the impact and the contribution differ and thus, the perturbation method is not able to quantify the contribution anymore (see also Grewe et al., 2010). We reworded the text to make this point clearer.*

Page 2, line 43 – Replace use of "relations" here and in other parts of the manuscript with "relationship"
*Thank you, we have modified the manuscript accordingly.*

Page 2 ,line 44-46 – Why does this sentence talk about "only the impact of road traffic emissions on O₃", I think this is what you want? Why does changing emissions only from road traffic emissions effect O₃ production from other sectors? Should they have not been kept the same in these two different simulations? Please make this description clearer.
*We would like to quantify the impact as well as the contribution of road traffic emissions to ozone. Changes in one emission sector affects ozone production in general and also ozone production by other emission sectors despite unchanged emissions in these sectors. For example, if NOₓ of road traffic emissions is reduced, this might actually increase the total ozone production in summer (see*

*details in Grewe et al. 2012) and increase ozone from other sectors, e.g. industry. As $NO_x$ from industry can "replace" $NO_x$ from road traffic, it can produce ozone more efficiently which is then attributed to the sector industry. Thus, the perturbation method implicitly alters the ozone production from other sectors. And we agree with the reviewer that they should be kept the same. This is provided by the tagging method where the ozone production from other sectors is largely unchanged. We modified the text accordingly.*

Page 2, line 44 – Change "As a variation of ..." to "As changes in …"
*We changed the text.*

Page 2, line 45 – Change "of" to "from"
*We modified the text accordingly.*

Page 2, lines 48-50 – I see the benefit of tagging but I think the distinction between tagging and perturbation experiments need to be made clearer at this point. Also how does the "impact" differ from "total effect". The use of language needs to be made clearer at this point and consistent throughout the manuscript.
*We modified the text and added more explanation to better explain the difference between the terms impact and contribution.*

Page 2, lines 52 – I would expect to read this statement earlier. Also there has been no mention about the impacts of $O_3$ on health and the environment until this point.
*We added a description on the impact of ozone on health and the environment earlier in the introduction.*

Page 2, Lines 53 – reword to "… reduce road traffic emissions to minimise their effect on climate"
*We changed the sentence as the reviewer suggested. Thank you.*

Page 3, line 63 -64 – move "on climate" from end of the sentence to after "$O_3$ effect"
*We changed the text.*

Page 4, line 79 – Is Figure 1 that useful? I understand that it is trying to convey the concept of TransClim but seems to lack useful detail and also other figures (2 and 4) seem to be showing a similar concept.
*We deleted figure 1 and 4.*

Page 4, lines 86-86 – All of the discussion so far has been on determining $O_3$ and its impact on climate. This is the first mention of other variables so could the above paragraph be talked of in more general terms or explain how the same method for $O_3$ can be applied to other chemical species.
*We rephrased the paragraph.*

Page 4, Section 2.2 – This is not a just a list of requirements but also what the algorithm does and also some results presented at the end. This needs to be sorted out to determine what you want to say here and this section reworded accordingly.
*We have formulated requirements which are important for the performance of TransClim. We actually tested various methods in Rieger (2018) and the algorithm which performs the best is presented in this manuscript. Here we only refer to the results in terms of conclusions. We have reformulated the text to clarify this.*

Page 4, line 91 – such as what other climate effect?
*Other possible metrics describing the climate effect are, for example, the global warming potential or the surface temperature change. TransClim so far only determines the stratosphere-adjusted radiative forcing. This is discussed in the conclusion of the manuscript.*

Page 4, line 93 – Do you also include the non-linear impact of CH$_4$?

*The non-linear behaviour of the tropospheric ozone chemistry is regarded by the approach used in TransClim. Thus, the non-linear effects of road traffic emissions on OH and thus on methane lifetime are considered as well. We have integrated this in the most recent tagging method (Rieger et al. 2018), which concentrates on OH and HO$_2$ tagging. The effect of road traffic emissions on methane lifetime is evaluated and discussed in sect. 3.2.*

Page 4, line 94-95 – This again is a bit confusing referring to the total change in O$_3$ and also the contribution from Traffic. Won't just perturbing the emissions from road traffic give you both of these of these quantities together? Clearly explain the difference between O$_3$ and O$_{3tra}$.

*Just perturbing the road traffic emission determines only the impact on ozone but not the contribution. To obtain the contribution on ozone, the tagging method needs to be applied in the base and perturbed simulation. We modified the description of the perturbation and tagging method in the introduction. We further moved the description of the tagging method from the appendix to the method section. This will improve the comprehension of the perturbation and the tagging method.*

Page 4, line 97 – Isn't the non-linear chemistry included within TransClim rather than considered?

*Thank you, we changed the text accordingly.*

Page 4, line 98 – "Road traffic emissions from different regions are accounted for" not regarded

*We changed the text accordingly.*

Page 4, line 101-102 – reword sentence to talk about local and remote response.

*Yes, done.*

Page 4, line 105-106 – How would you take into account future changes in background O$_3$? Also would this include effects of CH$_4$ and surface temperature on O$_3$? If this is not being done in the current study why mention it here as a requirement. Wouldn't it be better to say what the current tool is able to do and then propose future improvements in a different section?

*Different ozone backgrounds could be regarded by extending the LUT with emission variation simulations for other base years. This would include the effects of methane and surface temperature changes. As the reviewer suggested, we removed this requirement and added this future improvement to the discussion in the conclusions.*

Page 4, lines 108-109 – This is not a requirement but a result.

*It is important that the algorithm used in TransClim is computational very efficient, i.e. it computes the climate effect very fast and the errors are very low. Otherwise, it makes no sense to develop a new response model and a global circulation model could be used. Thus, this point is an important requirement which TransClim needs to fulfil.*

Page 5, line 113 – In Figure 2 are the two response fields identical for the two different regions? If this is only a schematic then why not show actual results from two regions to give the reader an idea of how the tool actually works?

*Yes, in this schematic the two look-up tables are identical. Showing actual results would only change the values on the z-axis. We don't think that this will improve the comprehensibility of the figure. Thus, we would like to keep it as it is.*

Page 5 line 114-115 – I know the emission scaling factors are mentioned further in the manuscript but on first reading this I didn't know what these were or how they had been decided to generate the algorithm. I think this is another point where reordering the methods would be beneficial.

*As the reviewer suggested, we restructured the method section. The emission scaling factors are now defined in sect. 2.4.3.*

Page 5, line 118 – I think this equation needs a label if it is going to be kept. Also if $\Delta x$ is provided by the look up table directly then I am not sure this equation is necessary in its current form. I think it would be better just to describe this in words like you have and then include the meaning of $\Delta x$ in what is currently labelled as equation 1. Also it would be good to say where the basis of these look up tables have been derived from (e.g. the perturbation simulations).

*As suggested, we deleted this equation and included the definition of $\Delta x$ in equation (1). As we restructured the method section, it should now be clearer that the basis of the lookup-tables is the set of emission variation simulations performed with EMAC.*

Page 5, line 119 – Specify what variable x can be.

*Yes, done.*

Page 6, line 125 – I am a bit confused about what $x_{ref}$ is. Is this the background values onto which the response to road traffic emissions are added onto? Should this not also be on a per region or per grid box basis? Also is $x_{ref}$ not also included in the calculation of $\Delta x$? Why is it in both calculations or are these different values?

*$x^{ref}$ stands for the variable x from the EMAC reference simulation. In the EMAC reference simulation, all emission scaling factors in all emission regions are set to 1. $x^{ref}$ is independent from the emission regions, but depends on the grid boxes and whether global or tropospheric means are considered.*

Page 6, line 127-128 – This is not obvious how the regionally derived values can be used to derived changes in global mean values or across the whole troposphere. Further explanation if required here of how this is done.

*We added further explanations to make this clearer to the reader.*

Page 6, line 130 – surely your approach is applicable to other emission perturbations? It is just utilised here for road traffic emissions?

*Yes, as later mentioned in the conclusion, other emission source such as emissions from railways and shipping can be also considered with this approach. This would require performing addition emission variation simulations. However, this study only focuses on road traffic emissions.*

Page 6, line 135 – I think section 3 definitely needs to come earlier as reading section 2 first just left me with lots of unanswered questions.

*Done, we restructured the method section.*

Page 6, line 138 – "eleven source regions"?

*We would like to keep the expression "emission region".*

Page 6, line 151 – label emissions for relevant year e.g. high road traffic emissions in 2010

*We changed the text.*

Page 7, line 163-166 – What is the impact of using global model simulations at 2.8 x 2.8 degrees to try and understanding the effect from road traffic emissions which normally take place at much finer resolution? Do the use of emissions at this resolution also impact the effectiveness of the model simulations to capture model the response to road traffic emissions?

*Mertens et al. (2020) analysed the impact of model resolution and emission resolution. The findings show that on larger scales, which are the driving factors for climate, the results for different model as well as emission resolutions agree fairly well. The model should not be used for air quality and health impacts. Local and regional effects need to be represented in an improved way. Hence we think the model's resolution is sufficient to capture larger scale free troposphere and upper troposphere ozone, relevant for radiation effects.*

Page 7, line 166 – I am surprised to see these simulations are free-running. Does this not introduce a potential additional problem of meteorological variability by using multiple simulations of the same model? Has this been accounted for or the impact of it calculated?

*The meteorology is calculated in a free running mode. However, it is binary identical for all simulations. This allows a "normal" variability of the meteorology, while the variability of chemical perturbations is low. This is achieved by performing the simulations in the Quasi Chemistry Transport model (QCTM) mode for EMAC. By prescribing climatologies, it decouples the chemical processes from the dynamical ones. Simulations performed in the QCTM mode show a "normal" meteorological variability, while this does not affect the variability of the chemical perturbations. Otherwise, we agree with the referee that this would have caused severe problems (see also Figure 1 of Deckert et al. 2011).*

Page 7, line 168 – change "trop-" to "troposphere"
*Thank you, we changed the word.*

Page 8 - What do you mean by Tg(VOC)? Which VOCs? How is this different to Tg(C) presented next to it? Surely you only need one definition of VOC emissions unless there is something different?

*Tg(VOC) and Tg(C) differ only by a conversion factor (161/210). To not confuse the reader, we deleted the last column of the table.*

Page 8, line 172 – I am not sure I understand what you mean by prescribing climatologies for radiation and the hydrological cycle. How does this impact the results? Can perturbations in emissions not therefore impact the radiation or meteorology?

*As the QCTM mode for EMAC decouples the chemistry and the dynamics, the 'noise' of the chemical perturbation is significantly reduced while the natural meteorological variability is kept unchanged. This allows for quantifying the climate response of small perturbations. This method suppresses the feedback of the chemical perturbation on the radiation calculation. Thus, the radiative fluxes due to the chemical perturbation are computed in a separate step. This is explained later in the manuscript. We added further explanations to the paragraph.*

Page 8, line 178 – $O_3^{tra}$ is mentioned here but you have not said yet how you have calculated this or what exactly it is.
*We moved the description of the tagging method before this paragraph.*

Page 8, line 178-179 – Is this the $O_3$ field in the model from the road traffic emission perturbation experiments? Or is this just the $O_3$ radiative fluxes from switching on the chemistry scheme?

*Yes, it is the ozone field which is changed by varying road traffic emissions. We modified the text.*

Page 8, line 180-182 – What is the third call to the radiation scheme doing that is different to the second? I think some clarification of what the radiation scheme is actually calculating is each step would be useful so it is easier to follow the process. I am a little confused about what is the second and third steps.
*We restructured the paragraph so that it is now clearer for the reader.*

Page 8, line 183 – are these radiative fluxes equivalent to instantaneous radiative forcings e.g. excluding fast and slow feedbacks on climate through adjustments to clouds?

*The radiative fluxes of the 2nd and 3rd radiation calls are computed after the concept of stratosphere-adjusted radiative forcing, i.e. the stratospheric temperatures can adjust to the new radiative equilibrium, but the tropospheric variables and stratospheric dynamics are kept fixed (e.g. Dietmüller et al., 2016).*

Page 9, line 189 – Is this $O_3^{tra}$? Can you provide a very brief summary of what the tagging is doing and how it is different to the perturbation method here rather than just referring to the appendix?

*Yes, this is $O_3^{tra}$. We added further explanation of the tagging method to this paragraph.*

Page 9, Figure 4 - whilst I appreciate showing 3D variables on a 2D plain is hard I wonder if you could also mark up the emission factors used in some of the different simulations so it is easier to see what has been perturbed. It is hard to tell from Figure 4 what some of the values are that have been used for each of the factors. It looks like you have done some experiments to reduce emissions but it is hard to tell what values these take for each emission used. If the information presented on this figure could be made clearer then it would improve its use. Perhaps a table might be better?

*We deleted the figure from the manuscript. The exact values of the emission variation simulations are now presented in table 2 as the reviewer suggested.*

Page 9, line 198 – "separately" implies that you have performed one a time tests, but I do not think this is what you have done?

*To not confuse the reader, we deleted the word.*

Page 9, line 203-204 – If you are varying two emissions at the same time then you are not really separating out their individual impacts. Are there are non-linear effects that need to be considered and mentioned here? Does this impact the use of TransClim when for example a perturbation of NOx emissions is applied but the results are based on changes from NOx + CO.

*There are also emission variation simulations where only one emission scaling factor is varied, and the remaining two factors are kept constant (see table 2). Hence, the separate effect of e.g. only $NO_x$ emissions can be quantified. We rephrased the sentence.*

Page 10, Figure 5 – I think improving this figure along with the other changes to the methods section would help the understanding of how TransClim works. For example if radiative fluxes are calculated separately then does this need to go in parallel to the other variables on the workflow? Need to put in Δx earlier on work flow diagram. How are the radiative forcings calculated for O$_3$ and O$_{3tra}$ separately?

*We adapted the figure and the corresponding text.*

Page 11, line 210 – I found the end of this sentence confusing "it is necessary to define an emission scenario as well as a reference scenario to which the emission scenario is compared to." This is suggesting you are comparing the emissions scenario to itself?

*Thank you, we rephrased the sentence.*

Page 11, line 216 – Could defining the radiative flux due to O$_3$ as "flxn(O$_3$)" be a bit misleading and potentially confused with the actual flux of O$_3$ concentrations. Would it be better to label it as RadFlxn(O$_3$)?

*To keep the variable names short and simple, we would like to keep the naming "flxn(O$_3$)". To not get confused with the actual ozone flux, we explicitly mentioned the radiative flux whenever the variable "flxn(O$_3$)" is used.*

Page 11, lines 216 – 218 – Similar to other points above, I am a bit confused of the double use of $O_3$ here in the calculations e.g. subtracting flxn(O3 – O3tra) from flxn(O3). Please clarify this and in other parts of the manuscript.

> *We deleted this paragraph in this section as it is not necessary here. We described the procedure of calculation the radiative fluxes of $O_3^{tra}$ in more detail in section 2.3.1.*

Page 11, lines 219-220 – The LUTs are based on the emission variation simulations using EMAC so you could state this here. I assume this is for all variables apart from the radiative fluxes which are treated separately?

> *Done. LUTs for the radiative fluxes are also generated from the emission variation simulations with EMAC.*

Page 11, line 220 – number 8.110.080 needs correcting.

> *The number of 8.110.080 LUTs results from the multiplication of 11 emission regions with 90 levels with 64 latitudes with 128 longitudes: 11*90*64*128 = 8110080. We rephrased the sentence.*

Page 11, line 221 – This is the first mention of $CH_4$ lifetime which needs further explanation of how it is used as it won't be the same as other variables.

> *As the relevant information is not the methane lifetime, but the number of LUTs, we exchanged the methane lifetime with another suitable variable (tropospheric mean of ozone).*

Page 11, line 224 – linearly interpolated between what, the response of a variable to two different emissions?

> *We rephrased the sentence.*

Page 11, line 225 – Do the LUTs just give Δx? If so just say this and do refer to earlier. I do not think you need to define it again here.

> *Yes, we changed the text accordingly.*

Page 11, line 229-232 – I am confused about the calculation of radiative forcings. Surely having a reference and perturbation radiative flux with TransClim is sufficient to calculate a forcing? What is the EMAC reference scenario and how is it different to the other reference scenario mentioned? Also how is the additional stratospheric-adjusted forcing different to that defined in the previous sentence?

> *The EMAC reference simulation refers to the EMAC emission variation simulation when all emission scaling factors are set to 1. The reference scenario is necessary for a TransClim simulation to determine the radiative forcing of the defined emission scenario. To differentiate these two, we renamed the reference scenario and called it "control scenario". We further adapted the paragraph.*

Page 12, line 244 – I assume Table 2 shows the emission scaling factors applied in TransClim and also in EMAC for evaluation purposes? What about $CH_4$?

> *Yes, (now) Table 3 shows the emission scaling factors which are used for TransClim and EMAC simulations. These simulations are compared in Sect. 3.1 to evaluate the performance of TransClim. For these simulations, the tropospheric OH is evaluated in Figures 5-7. Tropospheric OH is used to determine the methane lifetime. The performance of TransClim in determining the methane lifetime is later compared in Sect. 3.2.*

Page 12, line 244-247 – I agree that this is a good test for TransClim but am disappointed not to see the performance evaluated over other regions (see major comment for more details).

> *Thank you, this is a good point. As suggested, we added more evaluation simulations for the regions North America, South America and Asia in sect. 3.1.2.*

Page 12, Table 2 – A question for these evaluation simulations is how is a LUT generated for Eastern Europe for 1.3 scaling of all emissions whereas the EMAC simulations used to build TransClim for this region (Table S4) only contain reductions? I am not sure how TransClim would work for this type of emission perturbation or similar perturbations across other regions.

*As mentioned above (major comments), the way of representing the emission variation simulation in table S4 was misleading. For each of the 11 emission regions, 21 emission variation simulations have been performed spanning an emission variation range between 0 % and 200 %. Thus, the scaling of the road traffic emissions in Eastern Europe of 1.3 is included in the range of the LUTs.*

Page 12, line 249 – State what $O_3$ this is.
*We added this information and rephrased the whole paragraph.*

Page 12, line 255 – Why is there an overestimate over the Mediterranean? Could this be due to the way the experiment in Table 2 has been setup with reduction in $O_3$ precursors over Southern Europe?

*The emission reduction in Europe was chosen in such a way to obtain a large range of emission variations. Not only the emission region "Southern Europe", also other regions show large emission reductions. The overestimation is caused by the superposition of the results computed by TransClim for the different emission regions in Europe.*

Page 13, line 263-264 – If the southern hemisphere response is not significant and due to very small perturbations then is it worth showing this on Figure 7? Or is it better to show absolute changes? Can you change this Figure to only show the Northern Hemisphere of Europe as in Figure 6 where you would expect the effects of changes in European road traffic emissions to be more important. Also is the tropospheric $O_3$ response on Figure 7 the same as on Figure 6, is so why show it twice?

*We would like to keep the relative changes as this helps to better estimate the magnitude of the presented deviations. For consistency reasons, we would further like to show the whole world as the same data is shown again in the new Fig. 7 (comparison of emission variations in different emission zones). This helps to understand the larger deviations of the contributions shown in Fig. 7.*
*As suggested, we removed Fig. 6. Still, we think that the reviewer's point is important and hence we added a figure with the absolute values of all regarded variables in the appendix.*

Page 13, line 268-269- This is just for one emission and one region? Are these errors consistent over different emissions and source regions?

*As mentioned in the text, this particular example is just for $NO_x$ emissions over the emission region Germany. As suggested, we added further evaluations for different emission regions and for different strength of emission scaling to quantify the performance of TransClim.*

Page 13, line 270-276 – Why mention the interpolation and show Figure 8 here? Would this not be better in the methods section instead of the results section? Also I found that stating the interpolation errors are small is a bit misleading, especially when referring to Figure 8. For example the error gets worse if you go from say 1.0 to 3.0 or say 1.0 to 0.0. So it appears that for some emission perturbations errors will increase? I am not sure these are so small (see major comment).

*As suggested, we moved the discussion about the interpolation error and Fig. 8 to the section "Model description of TransClim". We further performed more evaluation simulations which all show that the errors caused by the applied algorithm in TransClim remain below 10 %.*
*The sketch (shown in Fig. 3, former Fig. 8) contains data points at 0, 1, 2 and 3 which are all used by the interpolation algorithm. So if you go from 1 to 3 for example, the data point 2 is used for the interpolation as well which will*

*significantly reduce the error. Actually, the error shown in the sketch is the largest error you can possibly get for this example.*

Page 14, line 277 – Underestimation of what?
*We rephrased the whole paragraph.*

Page 14, line 278 – TransClim reproduces EMAC results for emission perturbation over this one region
*We rephrased the whole paragraph.*

Page 15, line 287-288 – move "The transport demand was determined" to the front of the sentence
*Thank you, we changed the text accordingly.*

Page 15, line 288-289 – change sentence to "Emissions were generated from road traffic, railways and inland shipping as well as passenger and freight transport."
*Thank you, we modified the sentence.*

Page 15, line 292-293 – What do you mean by baseline scenarios? Is 2008 the baseline and 2020 and 2030 the future perturbation scenarios?
*Thank you. In the project VEU1, different emission scenarios were created. But in this manuscript, we consider only the baseline emission scenario. To not confuse the reader, we omit the word "baseline" in the sentence.*

Page 15, line 294 – What is the set up for EMAC?
*The model setup is described in Gottschaldt et al. (2013) and Hendricks et al. (2018). We refer to these papers at the end of the paragraph.*

Page 15, line 297-298 - How does future emissions in other European countries evolve? You have said how German emissions change but it is not clear what happened for other countries that will clearly have a different mix and magnitude of transport emissions.
*In the project VEU1, in order to determine the climate effect of German road traffic emissions, not the German road traffic emissions but the whole European road traffic emissions are neglected. This method of course assumes that the mix and magnitude of road traffic emissions would be similar in Germany and Europe. Hendricks et al. (2018) discuss this issue: "Estimating the German effect by this scaling procedure requires the assumption that the radiative forcing per emitted amount of pollutant is similar for the European and the German emissions. Since pollutants released over Europe usually experience vigorous mixing, uncertainties due to this assumption are probably small."*

Page 16, line 299-301 – How is the radiative forcing downscaled? Also how are the $CH_4$ lifetime changes determined?
*The radiative forcings are downscaled according to the ratio of German to European traffic emissions of $NO_x$. The methane lifetime change is deduced from the transport-induced changes on OH. The details of the calculations can be found in Gottschaldt et al. (2013). We added this information to the text.*

Page 16, line 306 – How have the emission scaling factors for other regions of Europe been calculated and are these for a particular year?
*The emission scaling factors bases on the European emission inventory for the year 2030 generated for the project VEU1. We added this information to the manuscript.*

Page 16, line 306 – Is NOx a combination of NO + NO2 here?
*Yes, we added this information to the text.*

Page 16, line 315 – replace "found" with "transported"
*Thank you, we replaced the word.*

Page 16, line 317-318 – TransClim reproduces the concentrations very well compared to what?
*Thank you, we adapted the manuscript.*

Page 17, Figure 9 – Can you convert some of these concentrations into more useful units e.g. ppb
*In Figure 8 (now) we compare the results of TransClim with the ones published in Figure 6 of Hendricks et al. (2018). As in this publication the SI unit mol mol$^{-1}$ is used, we would like to keep it to better compare the both results.*

Page 17, line 320-321 – Yes the emission inventory can make a difference but what about meteorological differences for these future scenarios. Could this also make an impact on the response of the variables in Figure 9?
*The simulations performed with TransClim and in the project VEU1 use the QCTM mode of EMAC. This mode decouples the chemistry and the dynamics by prescribing climatologies. In these QCTM simulations, the climate effect of the introduced perturbation has only a very small interannual variability. This implies that the chemical response of the perturbation relies only a little on the meteorological variations of different years (Hendricks et al. 2018).*

Page 18, line 323 – Which years?
*We changed the text accordingly.*

Page 18, line 325 – Change start of sentence to "TransClim also determines …"
*Thank you, we modified the text.*

Page 18, Figure 10 – NOx changes not shown on figure
*Thank you, we changed it.*

Page 19, line 334 – How much does German road traffic emissions increase by?
*We adapted the text.*

Page 19, line 335 – Change sentence to "A decrease in OH occurs in the free troposphere due to German road traffic emissions."
*We modified the text.*

Page 19, line 336 – Increase in OH due to increase in NOx?
*In the lowermost atmospheric layers, the increase of NO$_x$ causes a decrease of OH. Above, the OH concentration increases. We reworded the sentence.*

Page 19, line 348-349 – What do you mean by downscaling here?
*We explained the method here again.*

Page 19, line 349-351 – Are these global changes? Sentence Is a bit confusing so pleases reword.
*The values refer to the tropospheric mean. We restructured the sentence.*

Page 20, line 355 – is this not just a consequence of scaling the forcing by decreasing future traffic emissions?
*Yes, scaling the O$_3$ forcing with decreasing NO$_x$ traffic emissions decreases the forcing as well. However, it is not self-evident that TransClim obtains the same trend, as the tropospheric ozone changes non-linear with NO$_x$ emissions.*

Page 20, line 357-360 – How can the radiative forcing due to road traffic emissions be twice as large as the total? I think this needs more explaining and better descriptions of what you mean.

*The radiative forcing of the contribution of German road traffic emissions to ozone ($O_3^{tra}$) is twice as large as the radiative forcing of the total ozone change caused by German road traffic emissions. This is consistent with the findings in Grewe et al. (2017, their section 5.2) and Mertens et al. (2018, their section 4). We specified this in the manuscript.*

Page 20, line 269-270 – Yes the LUTs do account for the non-linear chemistry but the linear interpolation does introduce errors in this.

*Yes, the linear interpolation causes errors. The errors caused by the algorithm are discussed later in the section.*

Page 21, line 293 – replace "bases" with "based"

*We replaced the word.*

Page 21, line 293-294 – also mention tropospheric $O_3$

*This work focuses on tropospheric ozone. However, the LUTs contain also the data to determine stratospheric ozone. Thus, we would rather like to neglect the "tropospheric" to leave the statement more general.*

Page 21, line 297 – First time that it has been stated that emissions of 0-200% can be used in TransClim but not sure this has been tested thoroughly.

*We added further evaluations to the section "Model evaluation" where the variation of the emission scaling between 0 and 200 % is tested. It shows that the application of TransClim for a scaling between 0 and 200 % is justified.*

Page 21, line 211 – Change to "This enables the global and regional atmospheric response to be calculated."

*We exchanged the wording.*

Page 22, line 414 – change "base" to "are based"

*Yes, done.*

Page 22, line 415-416 – Also meteorological and transport patterns are an issue in different years. Why say that new LUTs need to be created when you use the current set of LUTs for different years and shown that they suitable? This statement is undermining the work presented in the manuscript.

*Thank you for this comment. We actually wanted to point out that the $O_3$ background concentration varies with time (here on decadal time scales and not day-by-day) and if a very different background concentration is considered, then a new set of LUTs need to be computed. Hence for future climates, the LUTs have to be recalculated. We modified the text to make it clearer.*

Page 22, line 420 – Change start of sentence to "It is easy to extend …"

*We changed the text.*

Page 22, line 424-425 – You mention traffic emissions and then quote studies on aviation emissions.

*This section shall give an overview over the response models which investigate the climate and air quality impact of traffic emissions. To our knowledge, so far only models for air traffic emissions exist.*

Page 22, line 436- replace "base" with "are based"

*We exchanged the word.*

Page 22, line 438-439 – What do you mean by "it does not regard the contribution of $O_3$ precursors" when referring to TM5-FASST? I think TM5-FASST does include the effect of $O_3$ precursors.

*As we mentioned in the introduction, in our manuscript "contribution" always refers to the contribution determined by the tagging method. To clarify this point, we modified the text accordingly.*

Page 22, line 442 – remove "reliably"

*We changed the text.*

Page 22, line 444 – change "regarded" to "included"

*We changed the wording.*

Page 22, line 444-445 – reword sentence

*Thank you for the hint. We restructured the sentence.*

---

## Referee Report (RR1)

**Comment on Revised manuscript "TransClim (v1.0): A chemistry-climate response model for assessing the effect of mitigation strategies for road traffic on ozone"**

The authors have made substantial efforts to thoroughly revise the manuscript taking into account the comments from both the reviewers. In particular, substantial effort has been made to amend the content and structure of the methods section, which improves the manuscript's readability. Furthermore additional evaluation of TransClim has been conducted on different source regions and for different emission scalings, which help document the performance of the model. I have made a few additional minor comments below for consideration by the authors (with line numbers relating to those in the revised non-track changed manuscript). Once these have been considered I am happy for the revised manuscript to be accepted for publication.

Throughout the revised manuscript I did a few sentences referring to TransClim assessing the climate effect of changes in road traffic emissions (section 2.1) and in other parts (e.g. section 2.3) specifically mentioning that TransClim assesses the impact on tropospheric $O_3$ and on climate via radiative forcing. For clarity it would be good to check throughout the manuscript and make consistent reference as to how TransClim assesses the impact on climate (i.e. via changes in tropospheric $O_3$ and stratosphere-adjusted radiative flux change at top of the atmosphere). For example can the link be made on Lines 98-100.

On Section 2.3 I am still wearing of calling them requirements. It is stated in the revised manuscript that the requirements were set out in Rieger (2018) where further testing of the algorithms took place. Also the sentence before the bullet points states "Here, we summarize the resulting key points for the final algorithm of TransClim". Therefore is seems to me that section 2.3 is more like objectives of TransClim or even processes included within the model.

In figure captions use schematic instead of sketch.

Line 167 - change end of sentence to "enables quantification of the climate response to a small perturbation."

Line 179 to 183 – In both bullet point 2 the calculation is referred to as "the stratosphere-adjusted radiative fluxes of the perturbed O3 field". Is point 2 stating the total O3 change from emission perturbations and point 3 is the difference between the total O3 and the O3 from traffic only. Can you just make clear what point 2 is (total O3 from all road emission perturbations?)

Line 209 – change fix to "fixed"

Line 230-231 – Consider revising to "Rieger, (2018) tested several different algorithms and the one that produced the best results is used in TransClim and described here".

Line 279-280 – "… the change of the variable x towards the EMAC reference simulation.". Should this be rephrased to say "… the change of the variable x with respect to the EMAC reference simulation."?

Line 283 – 284 – Similar to above. Change "… at top of the atmosphere of the emission scenario towards the control scenario …" to "… at top of the atmosphere for the emission scenario with respect to the control scenario …"

On Fig 4 and in section 2.6 it is mentioned that the algorithm is applied in each grid box (b) for each emission region (i). Would it be better to mention that the algorithm is applied on grid box in section 2.5 as well so that there is consistency between the sections?

Line 315 – don't need "regarded"

Line 332-330 – so similar problem as presented in Fig 5 but positive bias now?

Line 512-513 – Should this mention that the TransClim O3 response is based on simultaneous emission changes from all three precursors (NOx, CO and VOCs)?

Fig A1 – It is correct that there are negative values for Flxn(O3) at TOA? Is this including stratospheric O3 too?

---

## Author Response (AR2)

**Response to Reviewer #2**

*We would like to thank the reviewer for reading and commenting on our revised manuscript. We regarded all comments and incorporated them into our manuscript. Our reply is given below in italic.*

**Comment on Revised manuscript "TransClim (v1.0): A chemistry-climate response model for assessing the effect of mitigation strategies for road traffic on ozone"**

The authors have made substantial efforts to thoroughly revise the manuscript taking into account the comments from both the reviewers. In particular, substantial effort has been made to amend the content and structure of the methods section, which improves the manuscript's readability. Furthermore additional evaluation of TransClim has been conducted on different source regions and for different emission scalings, which help document the performance of the model. I have made a few additional minor comments below for consideration by the authors (with line numbers relating to those in the revised non-track changed manuscript). Once these have been considered I am happy for the revised manuscript to be accepted for publication.

Throughout the revised manuscript I did a few sentences referring to TransClim assessing the climate effect of changes in road traffic emissions (section 2.1) and in other parts (e.g. section 2.3) specifically mentioning that TransClim assesses the impact on tropospheric $O_3$ and on climate via radiative forcing. For clarity it would be good to check throughout the manuscript and make consistent reference as to how TransClim assesses the impact on climate (i.e. via changes in tropospheric $O_3$ and stratosphere-adjusted radiative flux change at top of the atmosphere). For example can the link be made on Lines 98-100.

> *Thank you very much for this comment. We checked the manuscript and specified the wording in the sections 2.1, 2.3 and 4.*

On Section 2.3 I am still wearing of calling them requirements. It is stated in the revised manuscript that the requirements were set out in Rieger (2018) where further testing of the algorithms took place. Also the sentence before the bullet points states "Here, we summarize the resulting key points for the final algorithm of TransClim". Therefore is seems to me that section 2.3 is more like objectives of TransClim or even processes included within the model.

> *As suggested by the reviewer, we changed the name "requirements" into "objectives" to better present the content of section 2.3.*

In figure captions use schematic instead of sketch.

> *Thank you, we changed the word sketch into schematic.*

Line 167 - change end of sentence to "enables quantification of the climate response to a small perturbation."

> *We changed the sentence as suggested by the reviewer.*

Line 179 to 183 – In both bullet point 2 the calculation is referred to as "the stratosphere-adjusted radiative fluxes of the perturbed O3 field". Is point 2 stating the total O3 change from emission perturbations and point 3 is the difference between the total O3 and the O3 from traffic only. Can you just make clear what point 2 is (total O3 from all road emission perturbations?)

> *Thank you. Point 2 refers to the $O_3$ concentration which is modified by the model chemistry. It includes changes in road traffic emissions. We adapted point 2 and point 3 to make this point clearer to the reader.*

Line 209 – change fix to "fixed"

*Thank you, we have changed the text accordingly.*

Line 230-231 – Consider revising to "Rieger, (2018) tested several different algorithms and the one that produced the best results is used in TransClim and described here".

*Thank you for suggesting revising the sentence. We modified the sentence.*

Line 279-280 – "… the change of the variable x towards the EMAC reference simulation.". Should this be rephrased to say "… the change of the variable x with respect to the EMAC reference simulation."?

*Thank you, we changed the sentence as suggested.*

Line 283 – 284 – Similar to above. Change "… at top of the atmosphere of the emission scenario towards the control scenario …" to "… at top of the atmosphere for the emission scenario with respect to the control scenario …"

*Thank you for this hint; we have modified the sentence.*

On Fig 4 and in section 2.6 it is mentioned that the algorithm is applied in each grid box (b) for each emission region (i). Would it be better to mention that the algorithm is applied on grid box in section 2.5 as well so that there is consistency between the sections?

*Thank you for this hint. To be consistent, we mentioned the application of the algorithm on the grid box as well in section 2.5.*

Line 315 – don't need "regarded"

*We omitted the word "regarded".*

Line 332-330 – so similar problem as presented in Fig 5 but positive bias now?

*Exactly, in this case TransClim overestimates the contributions $O_3^{tra}$, $OH^{tra}$ and $flxn(O_3^{tra})$ only in the Southern Hemisphere (see figure below). This is again caused by the small values in the Southern Hemisphere. To compute the relative differences, the absolute differences are divided by these small values which generate this noise.*

[Figure]

[Figure]

Line 512-513 – Should this mention that the TransClim O3 response is based on simultaneous emission changes from all three precursors (NOx, CO and VOCs)?

> *Yes, TransClim is based on simultaneous emission changes. We reworded the sentence to make this point clearer.*

Fig A1 – It is correct that there are negative values for Flxn(O3) at TOA? Is this including stratospheric O3 too?

> *The plot shows the net radiative fluxes due to ozone at TOA. It includes the tropospheric and stratospheric ozone. The net radiative fluxes are computed by adding the shortwave and longwave radiative fluxes (shown below) resulting into positive and negative radiative fluxes for ozone.*

**shortwave flux at TOA**

shortwave flux                                                    W/m**2

[Figure]

**longwave flux at TOA**

longwave flux                                                     W/m**2

[Figure]

**List of relevant changes**

- Specification of how TransClim assesses the climate effect in sect. 2.1, 2.3 and 4
- Change of term "requirements" to "objectives" in sect. 2.3 and sect. 4
- Extension of equation (5) in sect. 2.5: the fundamental equation of the algorithm also depends on the grid box b